# Proteomics, physiological, and biochemical analysis of cross tolerance mechanisms in response to heat and water stresses in soybean

Ramesh Katam[1]*, Sedigheh Shokri[1,2], Nitya Murthy[1,3], Shardendu K. Singh[4], Prashanth Suravajhala[5,6], Mudassar Nawaz Khan[7], Mahya Bahmani[8¤], Katsumi Sakata[9], Kambham Raja Reddy[4]*

1 Department of Biological Sciences, Florida A&M University, Tallahassee, Florida, United States of America, 2 Department of Horticulture Sciences, College of Agriculture, Tarbiat Modares University, Tehran, Iran, 3 Kentucky College of Optometry, University of Pikeville, Pikeville, Kentucky, United States of America, 4 Mississippi State University, Mississippi, Mississippi, United States of America, 5 Bioclues.org, Hyderabad, India, 6 Department of Biotechnology and Bioinformatics, Birla Institute of Scientific Research, Jaipur, India, 7 Institute of Biotechnology & Genetic Engineering, University of Agriculture, Peshawar, Pakistan, 8 Department of Agronomy and Plant Breeding, College of Agricultural Sciences & Engineering, University of Tehran, Tehran, Iran, 9 Department of Life Science and Informatics, Maebashi Institute of Technology, Maebashi City, Gunma, Japan

¤ Current address: Department of Biology, School of Science, Edith Cowan University, Joondalup, Perth, Western Australia, Australia
* Ramesh.katam@famu.edu (RK); krreddy@pss.msstate.edu (KRR)

**Data Availability Statement:** All relevant data are within the paper.

**Funding:** RK, and NM acknowledges the financial support from National Science Foundation Grant

## Abstract

Water stress (WS) and heat stress (HS) have a negative effect on soybean plant growth and crop productivity. Changes in the physiological characteristics, proteome, and specific metabolites investigated on molecular and cellular functions were studied in two soybean cultivars exposed to different heat and water stress conditions independently and in combination. Leaf protein composition was studied using 2-DE and complemented with MALDI TOF mass spectrometry. While the two cultivars displayed genetic variation in response to water and heat stress, thirty-nine proteins were significantly altered in their relative abundance in response to WS, HS and combined WS+HS in both cultivars. A majority of these proteins were involved in metabolism, response to heat and photosynthesis showing significant cross-tolerance mechanisms. This study revealed that MED37C, a probable mediator of RNA polymerase transcription II protein, has potential interacting partners in *Arabidopsis* and signified the marked impact of this on the PI-471938 cultivar. Elevated activities in antioxidant enzymes indicate that the PI-471938 cultivar can restore the oxidation levels and sustain the plant during the stress. The discovery of this plant's development of cross-stress tolerance could be used as a guide to foster ongoing genetic modifications in stress tolerance.

number 1156900 to carry out analysis of this research.

**Competing interests:** The authors have declared that no competing interests exist.

## Introduction

*Glycine max* (L.) Merr (soybean) is a legume that provides a significant source of proteins and fatty acids in both human and animal diets. It is an important legume crop grown for its combustion fuel, cooking oil, and protein with over 121.5 million hectares worldwide [1]. It is the largest source of feed protein in the world and the second-largest source of food oil [2, 3]. Soybean is widely adopted and cultivated crop across the climatic zones of the world. The crop plays a significant role in contributing to soil fertility as they are naturally capable of fixing atmospheric nitrogen and the root exudates of some legumes can solubilize phosphorus and other insoluble calcium-bound phosphorus compounds [4, 5]. The presence of legumes ameliorate the soil quality by encouraging microbial activity, especially around its rhizosphere, by contributing to organic matter restoration, and play a role in disease prevention and pest control [6].

Soybean has also been an important model crop for $C_3$ annual plants because of its strong response to climate change. For example, a 17% decrease in yield for 1˚C rise in temperature has been observed [7]. The overall production of soybean is severely limited by several abiotic factors that include flooding, drought, salinity, and acidity [8]. Due to its various developmental stages, these abiotic factors strongly impact the plant's growth. Therefore, it is essential to protect crop yields from higher and more frequent episodes of extremely high temperatures and drought both in current and future climates. Water or heat stress are involved in cell dehydration and affects various metabolic functions in plants. Water stress (WS) is one of the most debilitating factors of soybean crop with dehydration in plants, leading to a disruption in the water potential gradients, loss of turgor pressure, denaturation of proteins, leading to a lack of investigation in understanding the cellular membranes [9, 10].

Another devastating effect of dehydration is desiccation, in which the protoplasmic "free water" is lost, and the cell is required to survive on the water-bound within the cell matrix [11]. It has also been found that plants respond to dehydration by alternating levels of protein synthesis and protein degradation, with a recent evidence suggesting that there is a direct correlation between the accumulation of proteins synthesized by dehydration stress and the plant's physiological adaptations to water stress [12, 13]. Besides, when soybeans have encountered water stress during the reproductive stage, owing to a lack of plasticity to recover at this stage, there was a much more detrimental decrease in seed yields and its attributes, in comparison to the plants' grown under irrigated condition [14].

Heat stress (HS) in the form of high temperatures during flowering, is a cardinal factor limiting seed count in many crops, including soybean [15–18]. High temperatures are found in many southern regions of the United States during the germination season of soybean plants. Temperature above 30˚C affects germination by decreasing the seed vigor of the soybean. As a result, the levels of stachyose and phytic acid in soybean seeds are decreased, which leads to difficulties in membrane biogenesis and germination [19]. While it has been reported that exposure to elevated temperatures encourages oil and protein production in the soybean plants, extreme temperatures result in changes to the seed oil concentration particularly in the ratios of singular fatty acids to total fatty acids in the soybean oil [20]. High temperatures can also lead to desiccation of the seeds and cause abnormal exine structure during microsporogenesis resulting in pollen malformation [21]. Heat stress tolerance is controlled by adjustments in the membrane structure and function, tissue water content, protein composition, lipid activity, and primary and secondary metabolites [22]. Several reports showed changes in molecules in response to water and temperature stresses at transcription and protein levels that affect photosynthetic efficiency, and the activity of nitrate reductase. The levels of soluble proteins in soybean cultivars were directly correlated with the leaf rate of photosynthesis [23].

In response to drought, proteins involved in photosynthesis, signaling pathways, and reactive oxygen species detoxification were severely impacted [24]. Both heat and water stress in soybean induced overexpression of the *DREB1* gene family, several dehydrins, and LEA genes resulting in their over-representation among up-regulated genes in soybean plants under heat and drought stresses [25]. In soybean, most of the differentially abundant proteins are related to photosynthesis, ATP synthesis, and protein biosynthesis [26] in response to WS treatment. For instance, the cellular and biochemical components triggered by drought results in the activation or suppression of specific genes, and consequently the proteins involved in cell division, cell growth, and cell differentiation are affected [27]. In response to heat stress, overexpression of enzymes involved in homeostasis, as well as the accumulation of various chaperone proteins, especially heat-shock proteins, were observed [28]. The rhizomes of soybean plants have shown that dehydration has a negative effect on the levels of proteins that are responsible for protein transport and storage, ATP synthesis, metabolism, and signal transduction [29]. Therefore, proteomic analysis under multiple stresses will lead to better determination of the molecular pathways and the molecules associated with the complex cross-tolerance.

Earlier studies described the relative abundance in leaf protein composition under drought or heat stress conditions in various crop plants [26, 30]. However, the interactive effects and complex cross-tolerance mechanisms associated with physiological, biochemical, and proteome changes to heat and water stress are not well understood. As plants undergo a combination of multiple stresses in the field condition, they trigger defense mechanisms and cooperative systems, which under multiple stresses display cross-tolerance with each other to increase the plants' immune efficiency [31]. Cross-tolerance is a phenomenon in plants that makes them more tolerant to second stress after imposing under first stress, such as induction of stress memory after the stress [32]. Simultaneous occurrence of more than one stress can have both positive and negative impacts on the plants performance and adaptation [33]. Therefore, determining the key regulators that take part in orchestrated responses to concurrent stresses provide a better understanding of tolerance mechanisms. The objective of the present study was to assess the effect of water stress, heat stress, and combined stresses on the regulation of leaf proteins in two contrast soybean cultivars.

## Materials and methods

### Experimental facility

The study was conducted in four sunlit, Soil-Plant-Atmosphere-Research (SPAR) plant growth chambers located at Mississippi Agriculture and Forestry Experiment Station, Mississippi State, MS, USA (www.spar.msstate.edu). Chamber air temperature, $CO_2$, and soil watering were controlled to provide automatic acquisition and storage of the data from the units, monitoring SPAR environments every 10 seconds [34, 35]. The high temperature, 38/30˚, and drought stress at optimum and high temperatures were imposed by withholding irrigation until the soil water content reached 96% (-3.0 M Pa) of the control (-1.5 M Pa). Leaf samples were collected from the plants for proteome analysis.

### Plant growth and treatments

Two contrasting soybean cultivars, PI-471938 (slow wilting and high yielding) and R95–1705 (high in protein concentration and moderate yield potential) were used in the experiment. The detailed experimental procedure outline is shown in Fig 1. Soybean seeds for both cultivars were planted in three rows. Plants were thinned to 10 plants per row 12 days after emergence (DAE) and irrigated three times a day with half-strength Hoagland's nutrient solution [36]. The air temperature of 28/20˚C (day/night) was maintained until the beginning of the

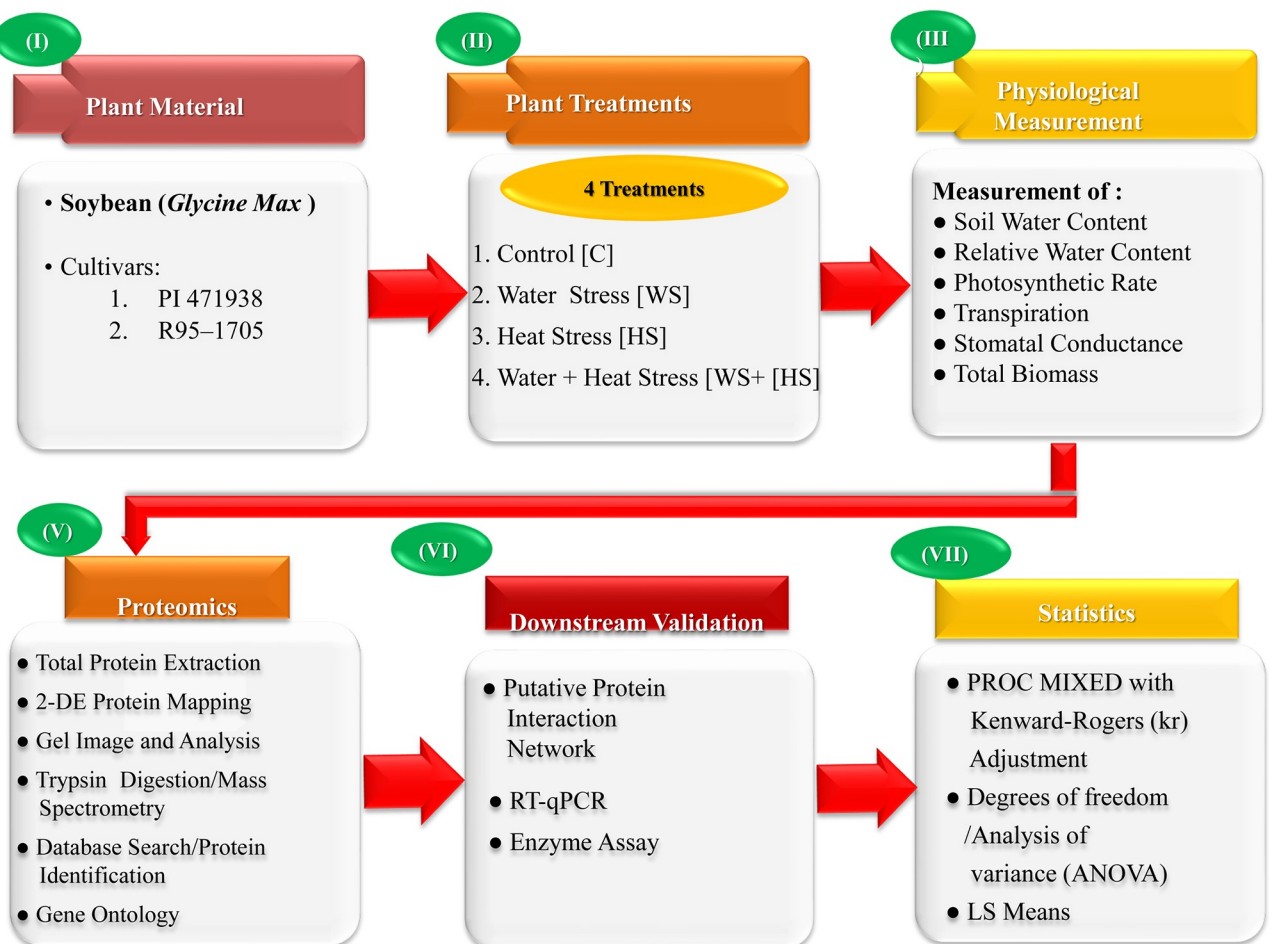

**Fig 1. Experimental outline, deciphering the analysis of physiological and biochemical changes in response to stress in soybean.**

treatments (30 DAE). After that, four treatments consisting of two levels of each factor, temperature (28/20° and 38/30°C) and irrigation (well-watered-WW, and water stress-WS), were imposed until the harvests (57 DAE). The control treatments consisted of 28/20°C and WW. The WS was imposed gradually as follows: no irrigation (30–31 DAE), watered 40% of the control (32–39 DAE), and no irrigation (40–50 DAE). The high temperature (38/30°) and drought stress at optimum and high temperatures were imposed by withholding irrigation until the soil water content reached 96% (-30 M Pa) of the control (-1.5 M Pa).

## Physiological measurements

**Soil water content.** The soil water content (SWC) was measured with a soil moisture probe (Delta-T Devices, Burwell, UK). Leaves of soybean plants were detached to measure the leaf's fresh, turgid and dry weights, and the relative leaf water content was determined as follows: LWC = (fresh weight–dry weight)/ (turgid weight–dry weight). Top most fully expanded leaves from six different plants were used for each treatment.

**Chlorophyll content.** Total chlorophyll was extracted by placing five 0.38 cm$^{-2}$ leaf disks for each row in a vial containing dimethyl sulfoxide (5ml) and incubated in the dark for 24h. After that, the absorbance of the supernatant was measured using a UV/VIS spectrophotometer

(Bio-Rad Laboratories, Hercules, CA, USA). The total chlorophyll was estimated and expressed on leaf area basis [37].

**Measurement of gas exchange and chlorophyll fluorescence.** These gas-exchange measurements were made on the uppermost fully expanded leaves (between 48 and 50 DAE) from six different individual plants in each treatment using a LI-6400 (LI-6400 photosynthesis meter, LI-COR Inc., Lincoln, NE) with an integrated fluorescence chamber head (LI-COR 6400–40 Leaf Chamber Fluorometer; LI-Cor Inc.). The temperature in the leaf cuvette was set to the daytime chamber air temperature and $[CO_2]$ was controlled by the $CO_2$ injection system to match the $[CO_2]$ treatments. The PAR provided by a 6400–02 LED light source was set to 1500 $\mu$mol m$^{-2}$ s$^{-1}$. Relative humidity inside the cuvette was maintained at approximately 50%. To measure fluorescence, the built-in leaf chamber fluorometer was used which uses two red LEDs (center wavelength about 630 nm) and a detector (sees radiation at 715 nm in the PSII fluorescence band). A flashlight ($>$7000 $\mu$mol m$^{-2}$ s$^{-1}$) achieved by using 27 red LEDs were used to measure the maximal fluorescence (Fm'). Rapid dark adaptation to measure minimal fluorescence (Fo') was achieved by turning off the actinic light while using the far-red LED (center wavelength at 740 nm). The far-red radiation drives photosystem-I (PSI) momentarily to help drain PSII of electrons. The gas exchange measurements such as photosynthesis (Pnet), stomatal conductance (*g*s), transpiration rate (Tr), internal carbon dioxide concentration (Ci), and chlorophyll fluorescence (Fv'/Fm') were in the analysis. For total biomass (TBM) measurements, 10 plants per treatments were used as replicates. All the plant-components were oven-dried inn a in a forced-air oven at 80°C for 48 h before weighing.

## Total protein extraction of soybean leaf

The uppermost fully expanded leaves were detached and placed in liquid nitrogen. Leaves were collected from six plants of the same cultivar then frozen in liquid nitrogen and stored at -80° C before protein extraction. Proteins were extracted following the modified procedure [38]. Briefly, frozen powder (6 g) was vortexed in 20 ml of 50 mM Tris HCl (pH 7.5) containing 2 M thiourea, 7 M urea, 2% Triton X-100, 1% DTT and 4% PVPP. The suspension was centrifuged at 5000 rpm and the protein was precipitated with TCA (15%). The protein pellets were washed twice in cold acetone (-20° C) and centrifuged for 15 min at 13000 rpm. Final pellets were resuspended in IEF rehydration solution [7 M urea, 2% CHAPS (w/v), 2 M thiourea, 0.2% DTT (w/v)] to measure the protein concentration [39].

## 2-DE protein mapping

An aliquot (300 $\mu$g in 100 $\mu$l) of the protein extract was loaded on to the tube gels and isoelectric focusing (IEF) was performed as described previously [40]. Tube gels were then loaded on a slab gel, and the proteins were resolved by electrophoresis.

## Gel image and analysis

Gels were scanned using a Gel Image system (Bio-Rad, Hercules, CA). The Analysis Set derived from three replicated gels of matched spots that were present on all the gels and the three replicated gels were analyzed using PD Quest (version 8.0.1). A one-way analysis of variance (ANOVA) was conducted to compare the mean protein spot densities and test if there was any difference in the protein spot abundance among the treatments and two cultivars studies. The differentially expressed spots (with P-values $<$0.05) showing significant differences were chosen for further analysis. Protein spots were manually excised from gels following in-gel digestion and MALDI/TOF mass spectrometry.

## In-gel trypsin digestion

The in-gel digestion mixture was a disulfide bond reduction. The resulting peptide mix was desalted with $C_{18}$ Zip Tips (Millipore), and 0.7 µl of the eluate and 5mg/ml matrix (α-cyano-4-hydroxycinnamic acid) was spotted on the ABI 01-192-6-AB MALDI plate (Applied Biosystems, Foster City, CA).

## Mass spectrometry, database search, and protein identification

Mass spectra were collected on the ABI 4700 Proteomics Analyzer (Applied Biosystems) MALDI/TOF mass spectrometer (MS), and protein identification was performed using the automated result dependent analysis of ABI GPS Explorer software, version 3.5 (Applied Biosystems). Data were analyzed as Peptide Mass Fingerprinting (PMF), and protein identifications were done by searching against the database using the MOWSE algorithm [41]. Both MS and MS/MS data were matched against Phytozyme the soybean taxonomic database. Only the proteins with a total score of confidence interval (C. I) % > 95% were considered as positive identities.

## Gene ontology (GO)

The identified proteins were mapped to Universal Protein Resource (UniProt KB) to assess their function as previously described [42]. The accessions were queried using batch Entrez to retrieve several sequences that mapped to different proteins. The annotations and accession numbers were retrieved using the GO Retriever tool and were grouped into different levels. Protein sequences were searched against gene ontology tools and the Target P program to derive functional classification, cellular localization and further validated using MapMan bin codes [43].

## Protein-protein interaction networks of differentially expressed proteins to WS+HS stresses

Time course expression data was used to estimate the interaction among the proteins in both cultivars. Protein-protein interactions (PPI) were estimated by temporal expression profiling utilizing an *S-system* differential equation [R1] as previously described [44]. Furthermore, from the association studies, 39 proteins then were interolog mapped to *Arabidopsis* database using GeneMania [45].

## Quantitative real-time polymerase chain reaction (RT-q PCR) analysis

Leaf samples (100 mg) collected from all stages of treatment were ground in liquid nitrogen. Total RNA was isolated using a modified CTAB-based protocol for RT-qPCR [46] and further purified using iScript cDNA Synthesis Kit (Bio-Rad, Hercules, CA, USA). A NanoDrop ND-1000 Spectrophotometer (Nanodrop Technologies, Wilmington, DE) and agarose gel electrophoresis were used to test RNA quality and quantity. Total RNA from each sample was reverse-transcribed using an iScript cDNA Synthesis kit (Bio-Rad, Hercules, CA, USA). Gene-specific primers were designed using NCBI database and the Primer Premier 5.0 as shown in S1 Table [47]. RT- qPCR was performed on Bio-Rad iCycler using the cDNA product (20 ng) in a 20 µl reaction mixture that includes 1 µL of forward and reverse primers of the corresponding transcripts, using SYBR® Green Universal mix (BioRad). PCR conditions were optimized for amplification of each gene before conducting relative quantitative experiment, using the specific primer pair and visualizing the PCR products by agarose gel electrophoresis. The PCR conditions were as follows: 95 ˚C for 30 s, then 45 cycles of 95 ˚C for 10 s and 60 ˚C for

30s. Data was acquired at 60 ˚C. Data was normalized using the actin gene Ct value and extent of change was calculated using the Ct value of the calibrator (control samples) using the formula 2-ΔΔCT. All assays for each gene were performed using five independent biological replicates for each gene in each treatment under identical conditions. Selected stress-responsive proteins (SRP); ascorbate peroxide, calreticulin, catalase, chalcone flavone isomerase, heat shock protein 70, peroxidase, peroxiredoxin, serine hydroxymethyl transferase 5, and superoxide dismutase were studied for transcriptional level expression analysis. The statistical significance of the results was evaluated with the Student's t-test ($p < 0.05$). All calculations were performed using Graphpad software V5.0.

## Enzyme assay

For all enzyme assays, leaf samples from control and treated plants were used in three replicates. To determine the enzyme activity, leaf samples were milled using a mortar and pestle. To determine the superoxide dismutase activity (SOD), the 3 mL reaction solution contained 13 mM methionine, 63 mM nitro blue tetrazolium chloride, 1.3 mM riboflavin, 50 mM phosphate buffer, and 50 mL of the enzyme extract [48]. The reaction mixture was incubated for 10 min and the absorbance was recorded at 560 nm. One unit of SOD activity corresponds to the amount of enzyme required for the inhibition of photochemical reduction of *p*-nitro blue tetrazolium chloride reduction by 50%. To determine the catalase (CA) activity, the 3 mL reaction solution contained 15 mM $H_2O_2$, 50 mM phosphate buffer (pH 7.0), and 50 mL of the enzyme extract [49]. The reaction was initiated by the addition of the 100 μL enzyme extract, and the decrease in absorbance of $H_2O_2$ at 240 nm for 30 s was recorded. For peroxidase (POD) activity, one gram of each leaf sample was separately milled in 5 mL of assay buffer. The homogenates were centrifuged at 12,000× *g* for 30 min at 4 ˚C [50]. Five mL of the assay buffer for the POD activity contained the following: 125 μM of phosphate buffer, 50 μM of pyrogallol, 50 mM of $H_2O_2$, pH 6.8, and one mL of the 20 times diluted enzyme extract. This was incubated for 5 min at 25 ˚C and, subsequently, the reaction was stopped by adding 0.5 mL of 5% (*v/v*) $H_2SO_4$. The amount of purpurogallin was determined by measuring the absorbance at 420 nm.

For ascorbate peroxidase enzyme (APX) activity, one gram of each sample was milled in 3 mL of extraction buffer 50 mM $KPO_4$ (pH 7.0), 2 mM ascorbate, and 5 mM EDTA at 4 ˚C [51]. The suspension was centrifuged for 20 min at 13,000× *g*. The supernatant was used for analyzing the enzyme activity. The reduction in the absorbance of APX indicates the activity within 1 min at 290 nm. One unit of APX activity was defined as the amount of enzyme required for catalyzing the oxidation of 1 mmol ascorbate per minute. The absorbance of non-enzymatic oxidation of ascorbate by $H_2O_2$ was used as control. Glutathione reductase (GR) was determined by measuring the reduction of GSSG by NADPH at 30 ˚C through the decrease in absorbance at 340 nm and via the extinction coefficient of 6.2 $mM^{-1}cm^{-1}$ measuring the absorbance at 340 nm [52]. The assay mixture contained 0.2 M potassium phosphate, 0.2 mM $Na_2EDTA$, 1.5 mM $MgCl_2$, 25 μM NADPH, 0.25 mM GSSH, pH 7.5, and 50 μL of enzyme extract in a 1 mL final volume. The reaction was initiated by the addition of NADPH.

## Statistical analysis

Statistical analyses were performed using SAS (SAS Enterprise Guide, 4.2, SAS Institute Inc., NC, USA). PROC MIXED with Kenward-Rogers (kr) adjustment of degrees of freedom was used for analysis of variance (ANOVA) to test the effect of treatments and cultivars, and their interactions on the plant and soil water status, chlorophyll concentrations, gas exchange and

fluorescence parameters, and total biomass. Treatments (temperature and irrigation) and cultivars were considered as the fixed effect, and individual measurements / rows were the random effects. The treatment comparisons were conducted by a least-square means (LSMEANS) procedure (at $\alpha$ = 0.05) and the letter grouping was obtained using pdmix800 macro [53].

## Results

Two soybean cultivars were used in this study to determine the changes in physiological, molecular, metabolite and enzyme activities in leaf tissues subjected to water stress or heat stress and combination of both.

### Physiological measurements

**Soil water content (SWC), chlorophyll content, measurement of gas exchange and chlorophyll fluorescence.** Water stress caused a severe decrease in soil water content (SWC) and relative leaf water content (LWC) leading to a reduction in photosynthetic rate ($P_{net}$), stomatal conductance ($g_s$) transpiration (Tr) and total biomass (TBM) in both cultivars. In plants grown at 28/20°C under well water (WW) conditions, the plant responses to water stress (WS) was more severe than high temperature (38/30°C) with the exception of the stomatal conductance ($g_s$) and the transpiration (Tr) which increased 33–39 and 71–91% at high temperature, respectively (Table 1). Among the physiological parameters, the C / T / IRR and the C / T interactions were significant ($P \leq 0.01$) for soil water content (SWC) and internal carbon dioxide concentration ($C_i$), where C, T, and IRR are cultivars, temperature and irrigation conditions respectively. The T / IRR interaction was significant ($P \leq 0.01$) for stomatal conductance

**Table 1. Temperature (T) and irrigation (IRR: well-watered, WW; water stressed, WS) effects on soil water content (SWC, $m^3\ m^{-3}$), leaf relative water content (LRWC, %), chlorophyll concentration (Chl, $\mu g\ cm^{-2}$), photosynthetic rate ($P_{net}$, $\mu mol\ CO_2\ m^{-2}\ s^{-1}$), stomatal conductance ($g_s$, $mol\ H_2O\ m^{-2}\ s^{-1}$), chlorophyll fluorescence (Fv′/Fm′), internal $CO_2$ concentration ($C_i$, $\mu mol\ CO_2\ mol^{-1}$), transpiration (Tr, $mmol\ H_2O\ m^{-2}\ s^{-1}$), and total biomass (TBM, g $plant^{-1}$) of two soybean cultivars (C) between 48 and 50 days after emergence.**

| Cultivar | Temperature | Irrigation | SWC | LRWC | Chl | $P_{net}$ | $g_s$ | Fv′/Fm′ | $C_i$ | Tr | TBM |
|---|---|---|---|---|---|---|---|---|---|---|---|
| PI 373819 | 28/20 °C | WW | 0.0855[bc] | 83.1[a] | 37.4[c] | 28.82[a] | 1.399[b] | 0.557[a] | 353[ab] | 16.11[c] | 14.48[a] |
| | 28/20 °C | WS | 0.0045[d] | 71.1[cd] | 39.6[bc] | 1.95[de] | 0.055[c] | 0.435[b] | 323[abc] | 1.66[d] | 10.28[abc] |
| | 38/30 °C | WW | 0.0899[ab] | 76.5[bc] | 39.5[bc] | 27.72[ab] | 1.866[a] | 0.543[a] | 362[a] | 27.51[b] | 13.47[ab] |
| | 38/30 °C | WS | 0.0013[d] | 64.2[e] | 39.6[bc] | 0.67[e] | 0.022[c] | 0.325[c] | 319[b] | 1.28[d] | 9.00[bc] |
| R95–1705 | 28/20 °C | WW | 0.0962[a] | 80.6[ab] | 40.0[abc] | 24.79[c] | 1.308[b] | 0.544[a] | 350[ab] | 16.81[c] | 13.33[ab] |
| | 28/20 °C | WS | 0.0033[d] | 70.1[d] | 39.5[bc] | 3.28[d] | 0.054[c] | 0.396[b] | 287[c] | 1.72[d] | 7.59[c] |
| | 38/30 °C | WW | 0.0829[c] | 79.5[ab] | 43.8[a] | 25.86[bc] | 1.810[a] | 0.551[a] | 360[a] | 32.08[a] | 13.86[a] |
| | 38/30 °C | WS | 0.0013[d] | 63.7[e] | 41.5[ab] | 2.60[de] | 0.015[c] | 0.331[c] | 193[d] | 0.84[d] | 6.79[c] |
| | *ANOVA* | *ANOVA* | | | | | | | | | |
| | | C | ns | ns | * | ns | ns | ns | *** | ns | ns |
| | | T | * | ** | ns | ns | * | *** | * | *** | ns |
| | | IRR | *** | *** | ns | *** | *** | *** | *** | *** | *** |
| | | C / T | ** | ns | ns | ns | ns | ns | * | ns | ns |
| | | C / IRR | ns | ns | ns | ** | ns | ns | *** | * | ns |
| | | T / IRR | ns | ns | ns | ns | ** | *** | ** | *** | ns |
| | | C / T / IRR | ** | ns | ns | ns | ns | ns | * | ns | ns |

Treatments (T, and IRR) were initiated 34 days after emergence. The data are the mean of the three-six individuals (the mean of ten individuals for total biomass).

Analysis of variance (*ANOVA*) between T, IRR, and C are given.

Significant at $^*P \leq 0.05$; $^{**}P \leq 0.01$; $^{***}P \leq 0.001$; and ns = non-significant (P > 0.05). Within columns for each experiment, means followed by same letters are not significantly different at $\alpha$ = 0.05.

($gs$), chlorophyll fluorescence (Fv'/Fm'), internal carbon dioxide concentration (Ci), and transpiration rate (Tr). whereas, C and IRR interaction was significant ($P \leq 0.05$) for photosynthesis (Pnet), internal carbon dioxide concentration (Ci), and transpiration rate (Tr). Both the treatments (temperature and irrigation condition, either WW or WS) significantly ($P \leq 0.001$) affected all the parameters except a few incidents of chlorophyll concentration, photosynthetic rate ($P_{net}$), and total biomass (TBM). Chlorophyll content was increased in response to combined water and temperature stress in the cultivar PI-471938 while it was reduced in the R95-1705 cultivar. With the exception of chlorophyll concentration in both cultivars, water stress in combination with high temperatures showed a greater reduction in these parameters. Water stress caused 14% reduction in leaf water content in both cultivars, while it was further reduced by 21% when both stresses were imposed. Compared to the control (well-watered), water stress severely reduced $P_{net}$ (>86%), $g_s$ (>95%), and transpiration (>89%), either alone or in combination with high temperature in both cultivars. However, high temperature alone increased both $g_s$ (>33%) and transpiration (>70%) in both cultivars. R95–1705 showed greater decrease in TBM (50%) than 471938 (38%) under water stress at high-temperature conditions.

## Analysis of stress responsive proteins

**2-D gel electrophoresis and identification of relative abundant stress-responsive proteins.** The 2-DE analytical gels of leaf proteins revealed that most of the proteins had a molecular weight ($M_r$) between 10 and 66 kDa, and pI between 4.3 and 7.9, a pattern typically observed in most of the leaf tissues (Fig 2). PD Quest digital image analysis and visual spot-by-spot validation of the match derived from 2-DE gels when carried out at a sensitivity reading of 5.0 revealed over 200 proteins in both the cultivars (S2 Table). Cultivar PI- 471938 showed 25% decrease to WS, an 8% increase to HS and a 10% decrease to both WS+HS in total proteins; while cultivar R95-1705 showed a decline by 6.6% to WS, 33% to HS, and a 10% to both WS+HS.

Comparative analysis of protein quantification profiles of both cultivars was carried out to identify the relative abundance of proteins in both cultivars to both stress conditions. Protein spots showing a ratio of at least >1.5-fold increase or <1.5 fold decrease between control and treatment was considered as threshold level to determine the relative protein abundance [54].

Comparative analysis of protein profiles revealed 39 protein spots, showing quantitative variation following water stress or heat stress that satisfies the 95% confidence interval (Table 2).

**Identification and functional classification of relative abundant stress responsive proteins.** All of these 39 protein spots accounted for 31 non-redundant proteins, with eight protein spots (4 proteins) showing similar protein accessions detected in multiple locations with differences in their isoelectric points and/or molecular weights on 2-DE gel. Relative protein abundance to various treatments between two cultivars and close up view of selected protein spots were is shown as in S1 and S2 Figs. To determine the functional categories related to stress responses, the proteins were classified into five major functional categories including metabolism (14), response to heat (7), photosynthesis (7), redox process (5), protein re-folding (3) and others (3) (Fig 3a). Over half of the proteins belonged to metabolism, response to heat and photosynthesis. The molecular function of each protein is shown in Fig 3b.

Some of the biological functions of the proteins in control and treatments of both cultivars were validated using MapMan (Fig 4; S3 Fig). The protein data were also analyzed to determine their association with individual organelles. The results predicted that they are localized in chloroplast, mitochondria, and others (Fig 5).

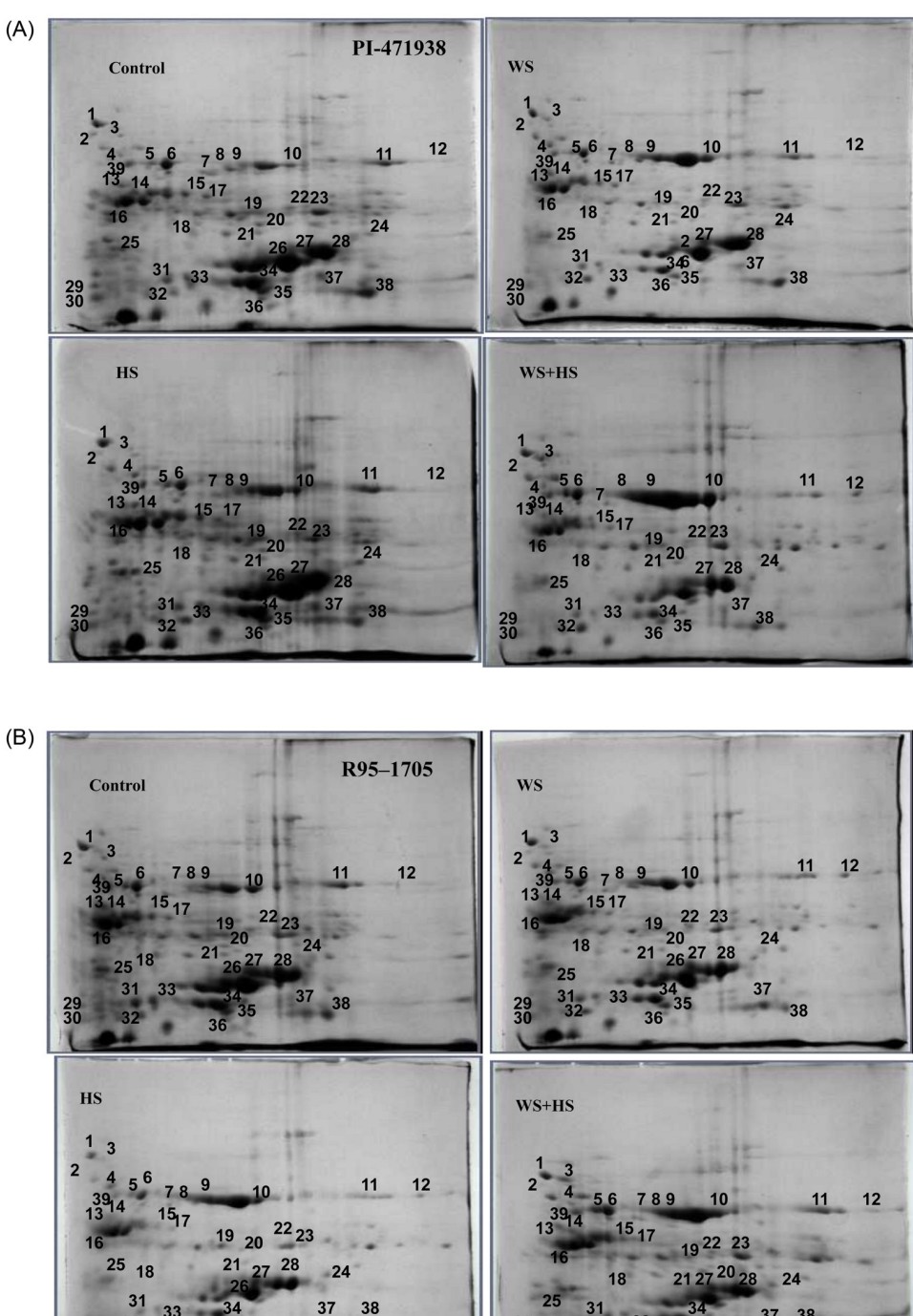

**Fig 2. Changes in leaf protein abundance (protein spots with numbers) to water (WS), heat (HS) and combined water and heat stress (WS+HS) in soybean cultivars PI-471938 and R95-1705.**

**Table 2. Identification of relatively abundant stress responsive proteins in soybean leaf proteome.**

| [a]Spot# | [b]Accession | [c]Phytozome ID | Description | [d]Mr (Da) /[e]pI-[f]Theo | [d]Mr (Da) /[e]pI [g]Exp | Molecular Function | Biological Process | Mowse Score | [h]Cov. % |
|---|---|---|---|---|---|---|---|---|---|
| | *1. Response to Heat* | | | | | | | | |
| *1 | P26413 | Glyma17g08020 | Heat shock Protein 70 | 73.9/ 5.20 | 66/4.55 | ATPase activity, Heat shock protein binding | Stress-related, Protein refolding | 876 | 50 |
| *3 | P26413 | Glyma17g08020 | Heat shock Protein 70 | 73.9/ 5.21 | 66/4.63 | ATPase activity, Heat shock protein binding | Stress-related, Protein refolding | 876 | 50 |
| *27 | Q39818 | Glyma12g01580 | Heat shock Protein 22 (mitochondrial) | 22.0/ 6.24 | 24/6.8 | Calvin cycle, Rubisco interacting | Stress-related | 395 | 52 |
| *28 | Q39818 | Glyma12g01580 | Heat shock Protein 22 (mitochondrial) | 22.0/ 6.24 | 26.6/ 5.8 | Calvin cycle, Rubisco interacting | Stress-related | 395 | 52 |
| *31 | P04795 | Glyma14g06910 | Heat shock protein 17.6 kda class 1 | 17.6/ 5.69 | 15/5.65 | Protein self-association, Unfolded protein binding | Response to heat, Stress-related protein complex oligomerization, | 255 | 66 |
| *32 | P04795 | Glyma14g06910 | Heat shock protein 17.6 kda class 1 | 17.6/ 5.69 | 15/5.65 | Protein self-association, Unfolded protein binding | Stress-related | 255 | 66 |
| 34 | B4X941 | VIGUN *Vigna unguiculata* | 17.7 kda class 1 heat shock protein | 17.8/ 6.85 | 17/6.2 | NA | Stress-related | 978 | 82 |
| | *2. Protein Re-folding* | | | | | | | | |
| 2 | P08824 | Glyma12g08310 | Chaperonin subunit alpha 60 kda | 61.7/ 5.23 | 63/4.4 | ATP binding, Calvin cycle, Rubisco interacting | Protein refolding | 312 | 27 |
| 6 | P02581 | Glyma05g09290 | Actin | 41.9/ 5.31 | 51/5.4 | ATP binding | Signal transduction | 217 | 19 |
| 11 | A0A762 | Glyma10g28890 | Calreticulin | 48.2/4.4 | 50/7.3 | Unfolded protein binding | Signal transduction | 150 | 36 |
| | *3. Oidation-Reduction process* | | | | | | | | |
| 5 | B0M1A4 | Glyma06g02040 | Catalase | 55.2/6.5 | 55/5.35 | Catalase activity, Heme binding, Metal ion binding | Redox, Responsive to H2O2 | 65 | 29 |
| 14 | Q9ZT38 | Glyma04g41990 | Alcohol dehydrogenase | 41.1/ 6.32 | 40/5.1 | Oxidoreductase activity, Zin ion binding | Oxidation, reduction | 1340 | 78 |
| 19 | Q43758 | Glyma11g15680 | Ascorbate peroxidase | 27.1/5.5 | 34/6.2 | Heme binding, L-ascorbat peroxidase activity, Metal ion binding | Redox, Cellular response to oxidative stress | 328 | 57 |
| 26 | C6SZ56 | Glyma19g42890 | Superoxide dismutase | 21.5/ 6.28 | 21/6.4 | Metal ion binding, Superoxide dismutase activity | Redox | NA | NA |
| 29 | B3GV28 | Glyma07g09240 | Peroxiredoxin | 17.4/5.4 | 14/4.3 | Oxidoreductase activity | Cell Redox hemostasis | 266 | 41 |
| | *4. Metabolism* | | | | | | | | |
| 4 | A8IKE5 | Glyma02g04320 | Alanine aminotransferase 2 | 52.1/ 6.92 | 54/5.9 | Photorespiration, Pyridoxal phosphate bonding, Transaminase activity | Biosynthetic process | 226 | 23 |
| 7 | O82560 | Glyma14g39420 | Glutamine synthetase | 47.9/6.4 | 47/5.1 | ATP binding, Glutamate-ammonia ligase activity, Identical protein binding | Metabolism, Glutamine biosynthetic process | 192 | 37 |
| 12 | C6ZJZ0 | Glyma18g150000 | Serine hydroxy methyl transferase 5" | 57.1/ 8.13 | 59/7.65 | Glycine hydroxymethyl transerase activity | Metabolism | 165 | 24 |
| 13 | O23963 | Glyma05g02670 | Translation elongation factor | 52.3/ 6.21 | 42/4.9 | GTPase activity, GTP binding, Translation elongation factor activity | Metabolism | 303 | 27 |
| 15 | E5RPJ6 | Glyma05g27260 | Pyruvate dehydrogenase | 38.9/ 5.70 | 39/5.6 | Catalytic activity | TCA | 1443 | 36 |
| 18 | O81278 | Glyma05g01010 | NAD dependent malate dehydrogenase | 43.9/ 6.47 | 33/5.5 | L-malate dehydrogenase activity | TCA process, Malate metabolic process | 185 | 24 |
| 20 | Q38IW8 | Glyma15g04290 | Triosephosphate isomerase | 33.3/6.3 | 28/6.25 | Triose-phosphate isomerase activity | Metabolism, Glycolytic process | 107 | 14 |
| 21 | Q93XE6 | Glyma20g38560 | Chalcone flavone isomerase 1A | 23.3/ 6.23 | 26/6.15 | Flavonoids, chalcone isomerase activity | Secondary metabolism | 325 | 54 |

*(Continued)*

**Table 2.** (Continued)

| [a]Spot# | [b]Accession | [c]Phytozome ID | Description | [d]Mr (Da) /[e]pI-[f]Theo | [d]Mr (Da) /[e]pI [g]Exp | Molecular Function | Biological Process | Mowse Score | [h]Cov. % |
|---|---|---|---|---|---|---|---|---|---|
| *22 | O22443 | Glyma09g02590 | Peroxidase | 38.6/6.0 | 41/6.7 | Heme binding, Peroxidase activity, Metal ion binding | H2O2 catabolic process | 1463 | 37 |
| *23 | O22443 | Glyma09g02591 | Peroxidase | 39.1/8.45 | 40/7.1 | Peroxidase activity | Metabolism | 1463 | 37 |
| 24 | Q9XJ23 | Glyma12g01000 | Acid phosphatase | 29.2/8.75 | 29/7.1 | Acid phosphatase activity, Gluconeogenesis | Metabolism | 134 | 39 |
| 33 | Q9SWA8 | Glyma12g04701 | Glycine-rich RNA binding protein | 16.7/5.5 | 14/6.20 | RNA binding, Transcription regulation | Metabolism | 424 | 55 |
| 37 | Q8GV24 | Glyma07g3710 | Nucleoside diphosphate kinase | 16.5/6.3 | 15/6.55 | ATP binding, Nucleoside diphosphate kinase activity | Metabolism | 1067 | 62 |
| 39 | O82561 | Glyma14g39421 | Glutamine synthetase | 47.9/6.4 | 47/5.1 | Nitrogen metabolism, Glutamine synthetase | Metabolism | 192 | 37 |
| | **_5. Photosynthesis_** | | | | | | | | |
| 8 | D4N5G3 | Glyma11g34230 | Rubisco activase | 14.6/6.8 | 50/61 | ATPbinding, Ribulose-1,5-bisphosphate carboxylase/oxygenase activator activity | Photosynthesis, Calvin cycle | 327 | 68 |
| 9 | Q6RIB7 | Glyma19g37520 | Enolase | 47.9/5.49 | 53/6.15 | Acetyl-CoA C-acyltransferase activity, Magnesium ion binding, Phosphopyruvate hydratase activity | Glycolytic process | 208 | 34 |
| 16 | I1JJ05 | Glyma02g45190 | Oxygen-evolving enhancer protein 2 | 27.7/8.27 | 29/4.7 | Calcium ion binding | Photosynthesis | NA | NA |
| 17 | Q2IOH4 | Glyma06g18110 | Glyceraldehyde 3-phosphate dehydrogenase | 36.8/6.72 | 37/5.65 | NAD binding | Metabolism, Glucose meta glycolytic process | 1052 | 70 |
| 25 | Q39831 | Glyma05g25810 | Chlorophyll A/B-Binding Protein | 27.9/5.29 | 22/4.9 | Chlorophyll binding | Photosynthesis, Light harvesting | 68 | 19 |
| 30 | Q39832 | Glyma19g06370 | Ribulose bisphophate carboxylase small chain 1 | 14.6/6.8 | 12/4.3 | Calvin cycle activity, Ribulose-bisphosphate carboxylase activity | Photosynthesis, carbon fixation, Photorespiration | 327 | 68 |
| 38 | Q6RUF6 | Glyma14g01470 | Fructose bisphosphate aldolase | 38.6/7.1 | 13.5/7.1 | Calvin cycle, Fructose-bisphosphate aldolase activity | Photosynthesis, Glycolytic process | 587 | 46 |
| | **_6. Others_** | | | | | | | | |
| 10 | Q6RIB8 | Other | 5-hydroxy tryptamine receptor 4 | 47.9/5.50 | 53/6.5 | G protein-coupled serotonin receptor activity | Morphogenesis | 208 | 34 |
| 35 | P10743 | Glyma08g21410 | Stem 31 kda glycoprotein precursor | 29.4/6.7 | 15/6.2 | Nutrient reservoir activity, Acid and other phosphatases | Seed storage | 720 | 65 |
| 36 | Q9ZTZ2 | Glyma17g16620 | Late embryogenesis abundant protein | 49.5/7.1 | 12/6.1 | Embryo development ending in seed dormancy | Seed development | 789 | 40 |

[a] Spot number as given on the 2-D gel image in Fig 2.

[b] Protein identification number as in Uniprot/NCBI database.

[c] Protein identification number as in Soybean phytozyme.

Database.

[d] Protein molecular weight.

[e] pI value.

[f] Theoretical value.

[g] Experimental Value.

[h] Identified peptide coverage.

* Protein resolved in multiple spots on 2DE.

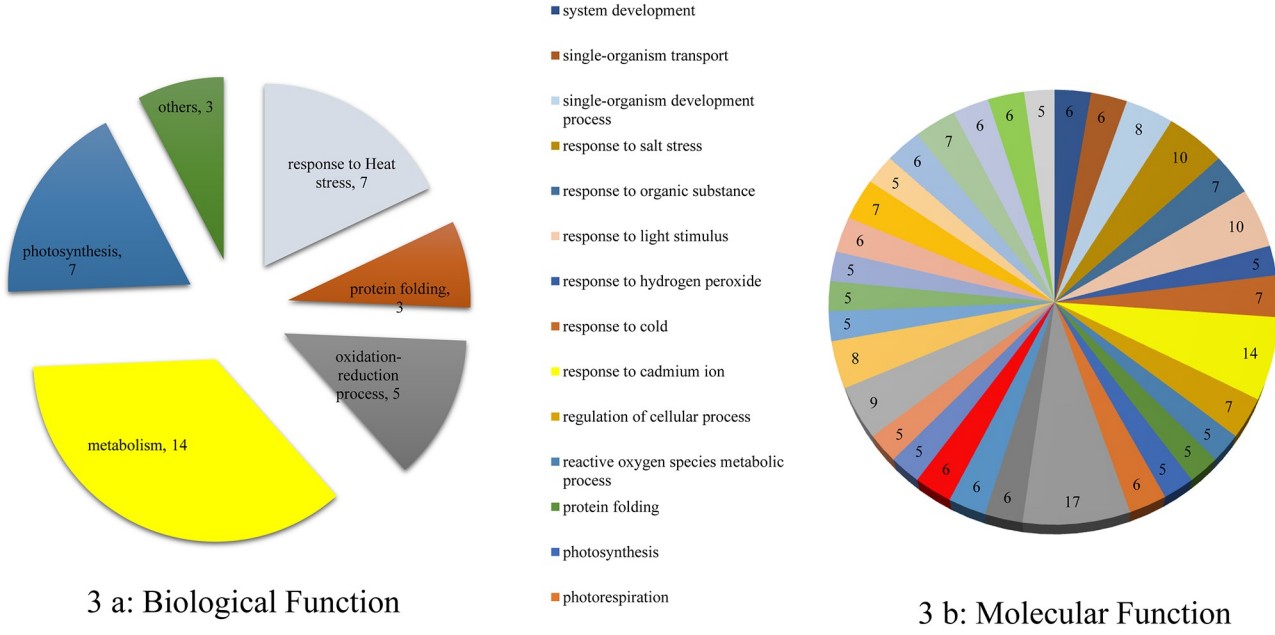

- system development
- single-organism transport
- single-organism development process
- response to salt stress
- response to organic substance
- response to light stimulus
- response to hydrogen peroxide
- response to cold
- response to cadmium ion
- regulation of cellular process
- reactive oxygen species metabolic process
- protein folding
- photosynthesis
- photorespiration

3 a: Biological Function

3 b: Molecular Function

**Fig 3. Gene ontology of stress responsive proteins: (a) Biological, (b) Molecular functions.**

**Clustering and dynamics of stress-responsive proteins.** Expression analysis of the thirty-nine relatively abundant proteins were carried and analyzed (Fig 6). Four clusters of protein expression were recognized in both cultivars.

In cultivar PI-471938 (slow wilting and high yielding) cluster I proteins that were increased in abundance to either water stress (WS) or heat stress (HS) or the combination of water and heat stress (WS+HS) were identified (Fig 6a). Out of eight proteins (# 3, 4, 9, 10, 12, 17, 24, and 27) the majority of them are involved in metabolism (3) photosynthesis (2), and response to heat (2) and other (1). Among the proteins in Cluster I, four proteins showed a high abundance to more than two stresses (#4 10, 17 and 27) of which, two proteins (#10 and 17) showed

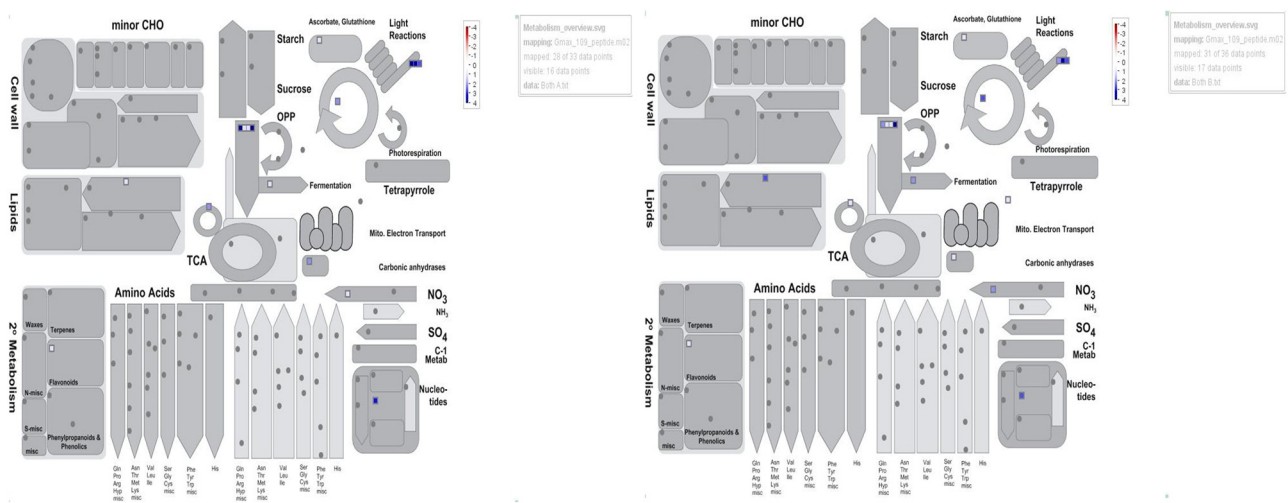

**Fig 4. Phytozyme confirmations of protein identities using MapMan.**

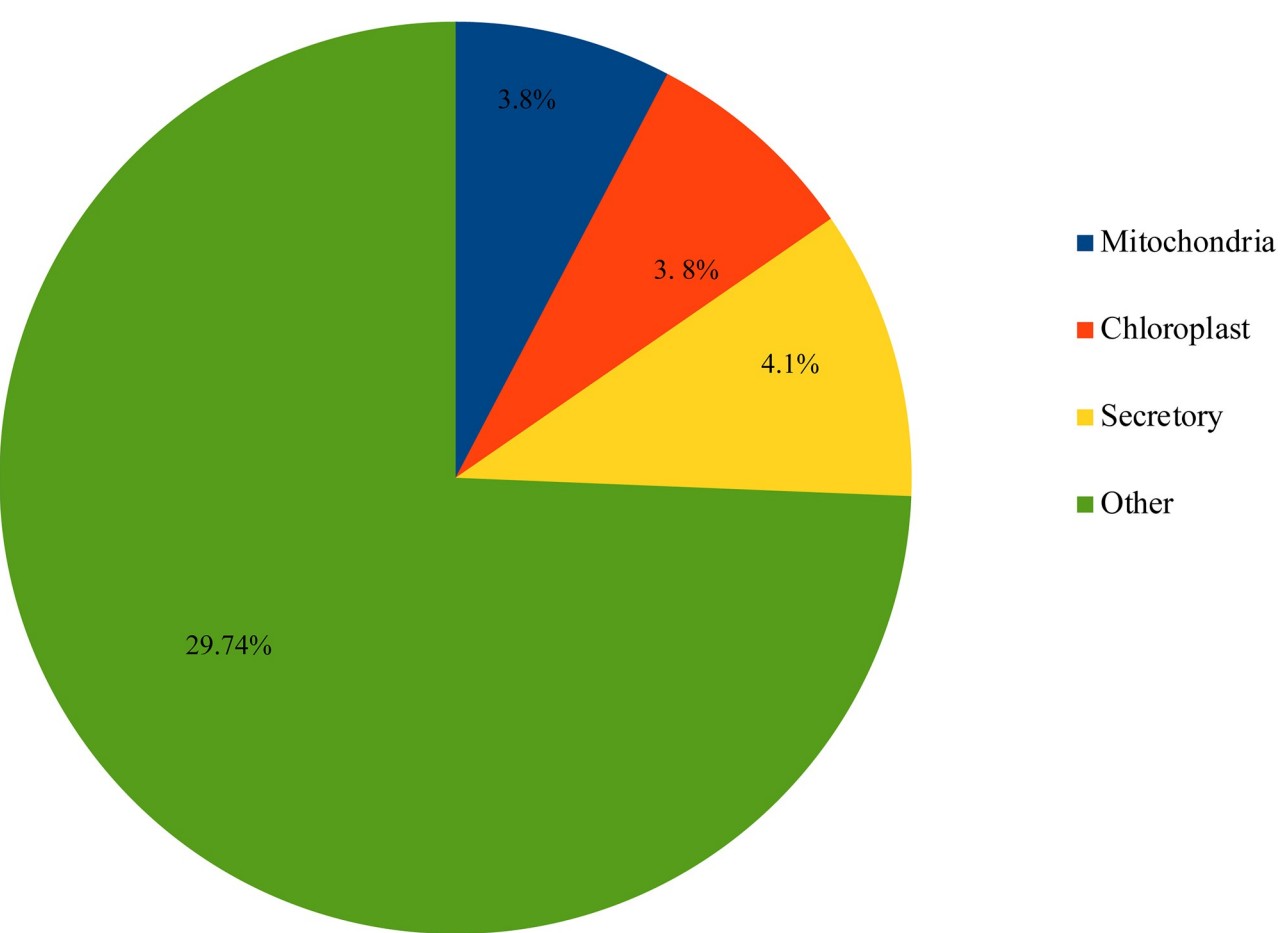

Subcellular localization of stress responsive proteins (SRPs)

**Fig 5. Sub-cellular localization of stress responsive proteins as predicted using TargetP.**

increase in abundance to all three types of stresses. Cluster II includes 19 proteins (#2, 5, 6, 7, 13, 14, 15, 19, 20, 21, 22, 23, 26, 29, 30, 34, 37, 38, 39) that are in low abundance in response to at least one or more of the three stress types. The majority of these proteins involved in metabolism (9), oxidation-reduction reactions (5). Fifteen proteins in cluster II were low in abundance to two or more stresses. Seven of them (#7, 13, 15, 21, 22, 29, 38) were low in abundance to all three types of stresses; the majority of them were involved in metabolism. None of the proteins fall under the category of Cluster III in PI-471938 cultivar. Cluster IV includes 12 proteins (#1, 8, 11, 16, 18, 25, 28, 31, 32, 33, 35, 36) exhibiting either low or high in abundances to each stress type/s, with the majority of them were involved in response to heat (3), and photosynthesis (3).

In cultivar R95-1705 (high in protein concentration and moderate yield potential), cluster I displayed a group of 10 proteins increased in their abundance to at least one type of stress, either to WS, or to HS, or the combined WS+HS (Fig 6b). Those proteins (#3, 4, 7, 8, 9, 12, 14, 23, 24, and 25) are mainly involved in metabolism (5), following photosynthesis (3), response to heat (1), and oxidation-reduction reactions (1). Among these proteins five of them showed resistance to two or more stresses involved in metabolism and photosynthesis. Four proteins showed high abundance to WS, and three to HS. Protein # 7 and 12 involved in metabolism

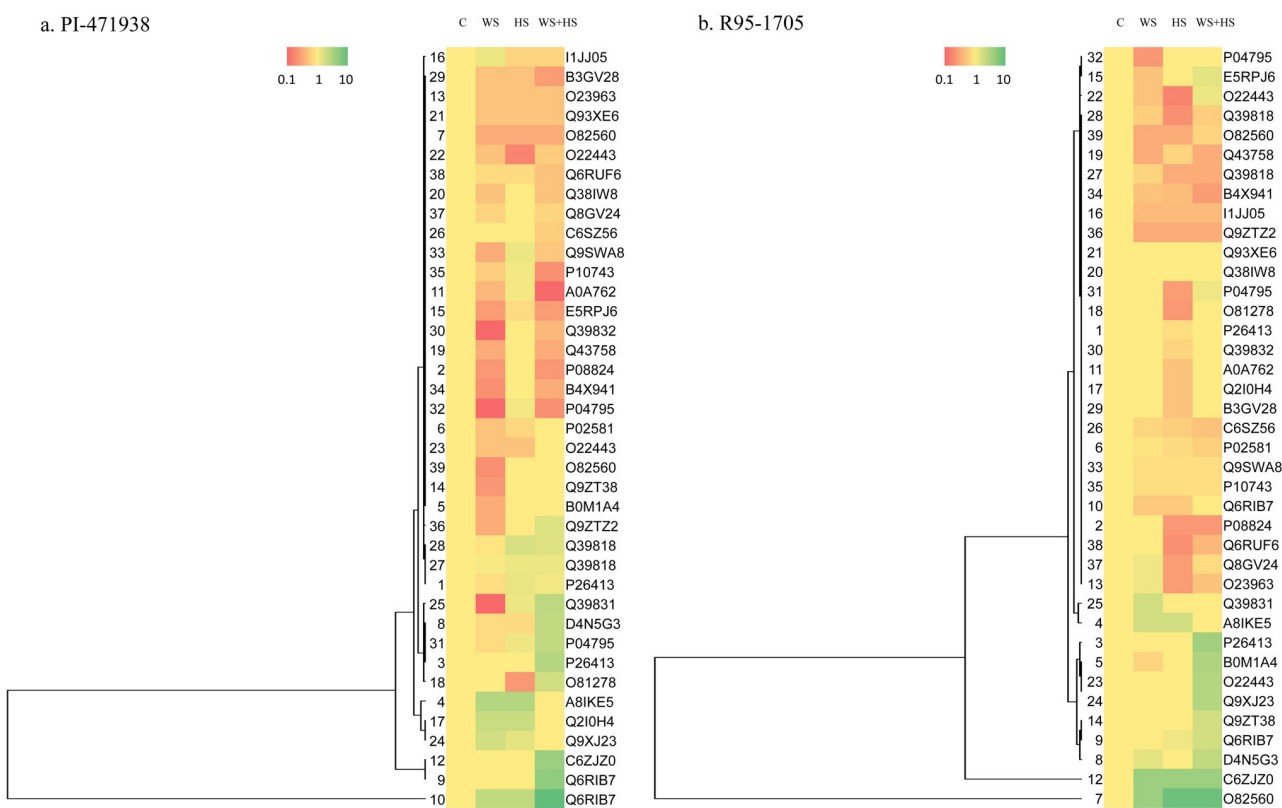

**Fig 6. Relative abundance of stress responsive proteins (SRPs) in a) PI-471938 and b) R95-1705 cultivars as depicted in hierarchical cluster.** The protein abundance ratios were used for cluster analysis by a hierarchical clustering method (centroid linkage Protein abundance ratios were used for cluster analysis by a hierarchical clustering method. Columns (from left to right): Control(C), Water Stress (WS), Heat Stress (HS), and Combination of Water and Heat Stress (WS+HS).

expressed in different quantities to all stresses studied. Cluster II includes 22 proteins that are in low abundance to one or more of the stresses (either WS, or HS or WS+HS) treatment (Spot #1, 2, 6, 10, 11, 16, 17, 18, 19, 22, 26, 27, 28, 29, 30, 32, 33, 34, 35, 36, 38, 39) and include those involved in response to heat (5), photosynthesis (4), metabolism (4), protein refolding (3), oxidation-reduction (3), and others (3). Among these, sixteen proteins showed reduction in abundance to two or more stresses, majorly involved in response to heat. ten proteins showed reduction in abundance to all three stresses. Cluster III proteins (spot # 20, and 21), which are involved in metabolism showed no response to water stress, heat stress, and combined WS+HS stresses. Cluster IV includes six proteins (Spot #5, 13, 15, 22, 31, and 37), mostly involved in metabolism (4) which showed mixed response to the stresses, low abundance to one type and high to another type stress.

## Putative PPI networks of differentially expressed proteins to abiotic stresses

Relative abundance of stress responsive-proteins were studied in time-course expression for the combined treatment of water and heat stress to determine the interactions among the proteins in both cultivars. Time course expression data was used to estimate the interaction among the proteins in both cultivars to understand the interaction levels of proteins to the biological function in response to both WS+HS stress over a 3-week duration. Thirty-two proteins have shown interaction in the PI-471938 cultivar, while 29 have interacted in R95-1705

cultivar. In PI-471938 cultivar, 95 promotive and 61 inhibitive interactions were observed, while 113 promotive and 139 inhibitive interactions were observed in R95-1705 cultivar (Fig 7; S3 Table).

To further validate the interaction, the proteins that showed high abundance to combined WS+HS were also analyzed to determine the frequency of promotive and inhibitory interactions in both cultivars. PI-471938 showed high abundance of 11 proteins belonging to heat resistance (3), oxidation-reduction (2), metabolism (5), and photosynthesis (2) to combined WS+HS (S4 Table). These proteins displayed 41 promotive and 15 inhibitive interactions. Among the promotive interactions, the majority were involved in metabolism, response to heat and photosynthesis. Cultivar R95–1705 showed a high abundance in 8 proteins involved in resistance to heat (4), photosynthesis (2), metabolism (1), and others (1) (S5 Table). These proteins displayed 28 promotive and 36 inhibitive interactions. Among the promotive interactions, majority were involved in metabolism.

When the 39 SRPs were subjected for interolog mapping and protein interaction analysis, heat shock protein 70 (MED37C) showing potential interactions with stress-related proteins indicate that these orthologs in *Arabidopsis thaliana* genome have good potential interacting partners (Fig 8).

## Comparative studies of mRNA transcript analysis with protein expression

Leaf samples from control and stressed plants were collected for mRNA extraction and then analyzed for RT qPCR. Results showed that, in cultivar PI-471938, the mRNA levels of ascorbate peroxide, chalcone flavone isomerase (CHI), serine hydroxymethyl transferase 5 (SHMT), calreticulin (CALR), peroxidase (POD), heat shock protein 70 (HSP 70), superoxide dismutase (SOD) showed up-regulation to all stress treatments, while catalase and peroxiredoxin showed reduced levels to HS treatment (Fig 9a). In R95-1705 cultivar, majority of the transcripts showed mixed responses (Fig 9b). Catalase and serine hydroxymethyl transferase 5 showed up-regulation of transcripts while calreticulin, heat shock protein 70, peroxidase, peroxiredoxin and superoxide dismutase showed down-regulation to all stress treatments. Expression of ascorbate peroxide increased to HS while reduced to WS, and WS+HS treatments. Chalcone flavone isomerase levels were increased to WS and HS while this transcript did not express at combined stress treatment.

## Determination of enzymes activities

The enzyme activities in both cultivars were measured for both control and stress treated plants in replicates. Under control conditions, relatively, the activity of superoxide dismutase (SOD; EC 1.14.1.1), peroxidase (POD EC.1.11.1.7), and catalase (CAT, EC 1.11.1.6), were low in PI-471938 cultivar compared to those in R95-1705, whereas ascorbate peroxide (APX; EC 1.11.1.11), glutathione reductase (GR; EC 1.8.1.7) showed higher activity in PI-471938 cultivar (Fig 10). The activity of SOD, POD, APX and CAT was increased in PI-471938 while it was reduced in R95-1705 cultivar when treated with WS, HS, and WS+HS. The activities of GR were higher in all three types of stresses in both cultivars, with an exception that GR has not been detected in R95-1705 cultivar when treated with combined stresses.

## Discussion

Plants' responses to concurrent stresses often occurring in the field are very exclusive, when compared to individual stress treatments and will display cross-tolerance to better adapt to those stresses [55]. We report the result of our studies on two contrast soybean cultivars to

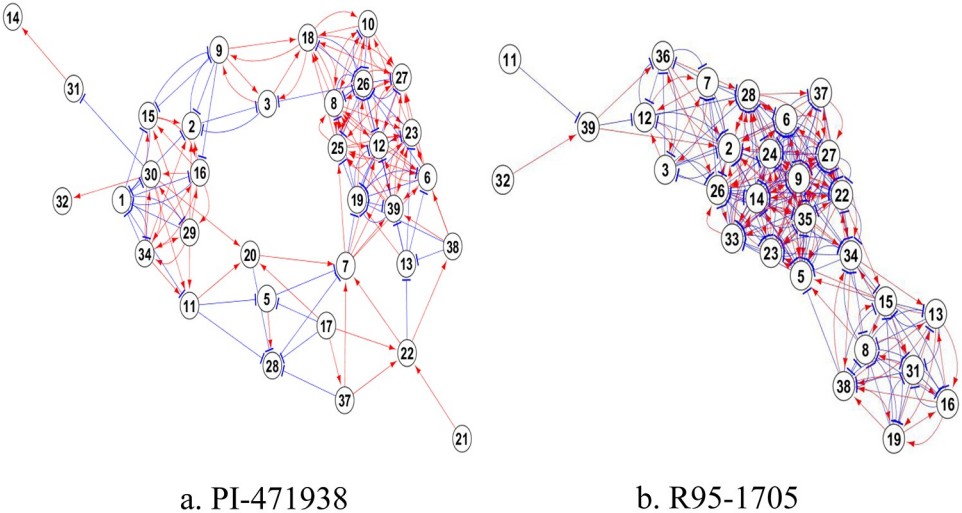

a. PI-471938                                b. R95-1705

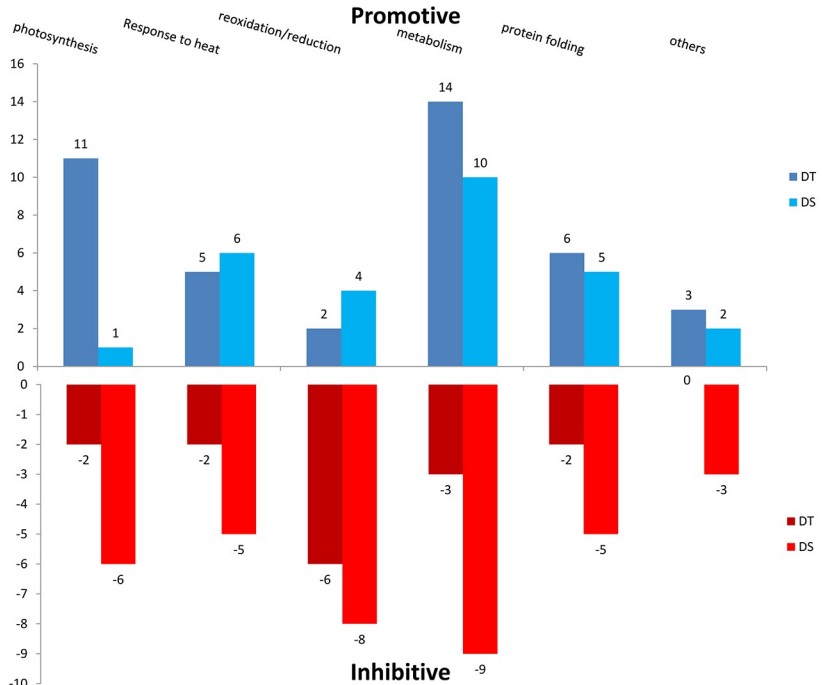

c. Promotive and inhibitive biological functions of both cultivars

**Fig 7. Protein-protein interactions among Stress Responsive Proteins (SRPs) in PI-471938 and R95-1705 cultivars in response to combined water and heat stresses.** Protein interactions in a) PI-471938 and b) R95-1705 cultivars. The number of interacting proteins involved in various biological functions is shown in c. The estimated interactions were evaluated based on a goodness-of-fit calculated from "multiple-correlation coefficient" ($R^2$) between an expression profile and simulated profile of a protein positioned on the downstream side of an interaction. The interactions showing an r2 value (coefficient of determination) >0.9 were considered as candidate interactions. We calculated the $R^2$ corresponding to the interaction that a protein regulated the expression of another protein based on a modified version of the S-system differential equation. Spot numbers are the same as in Table 2.

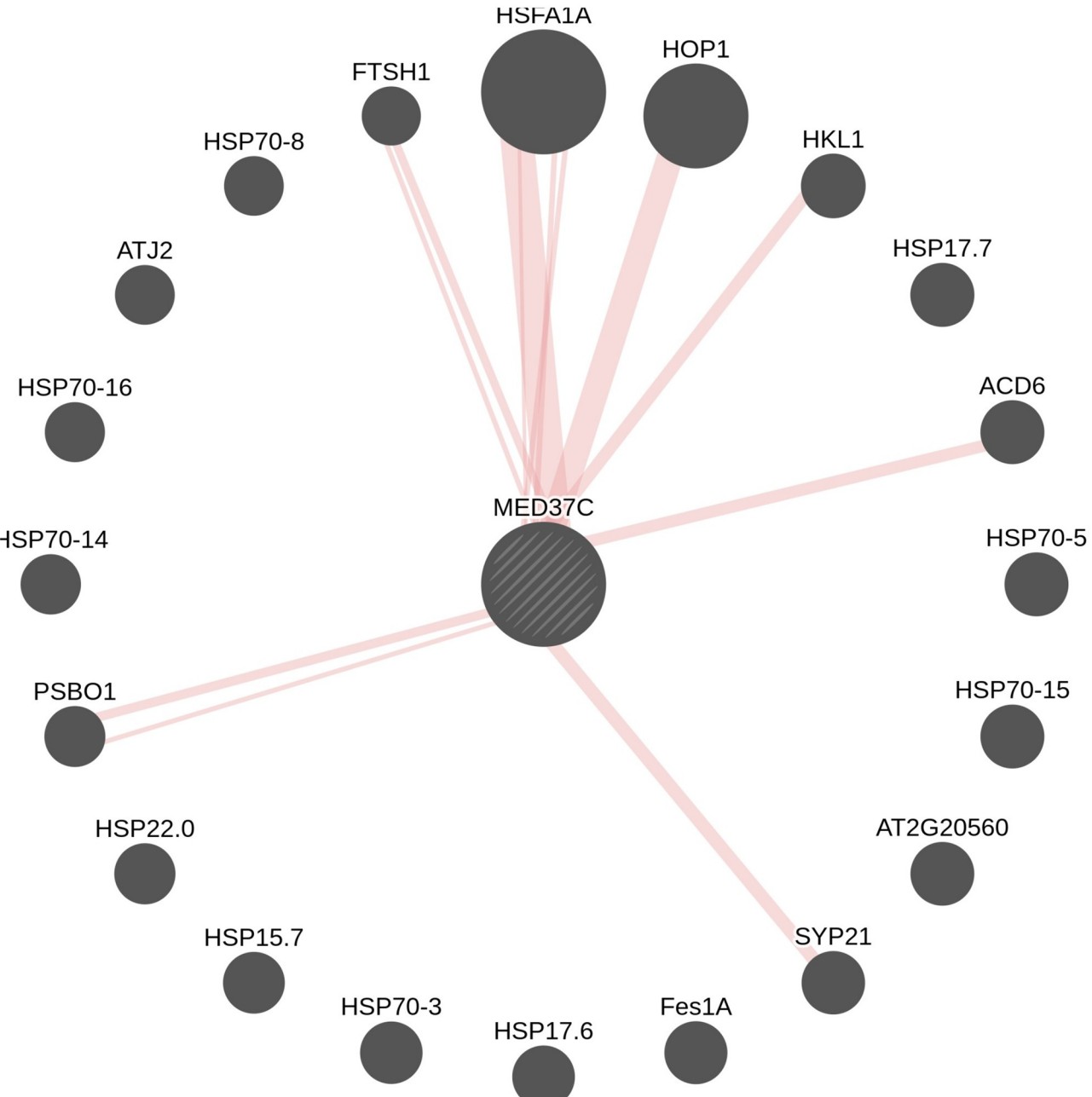

**Fig 8. Protein-protein interaction map of the gene MED37C.** Physical interactions shown in the form of edges (Thicker the edge, greater the number of experiments for which interactions were ascertained).

determine the physiological and molecular mechanism for cross-tolerance between multiple stresses.

## Greater reduction in total biomass to stress treatments in R95-1705 cultivar

The observed decrease in photosynthetic rate ($P_{net}$), stomatal conductance ($g_s$), transpiration (Tr) and total biomass (TBM) under water stress alone or in combination with high

(A)

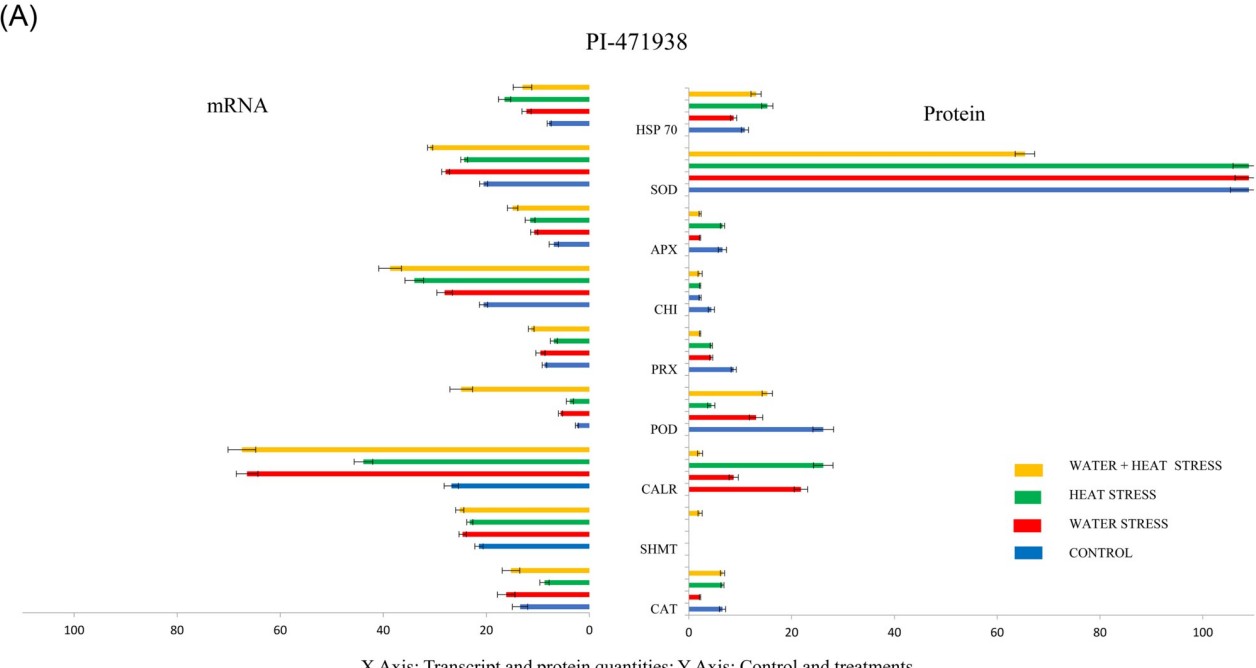

(B)

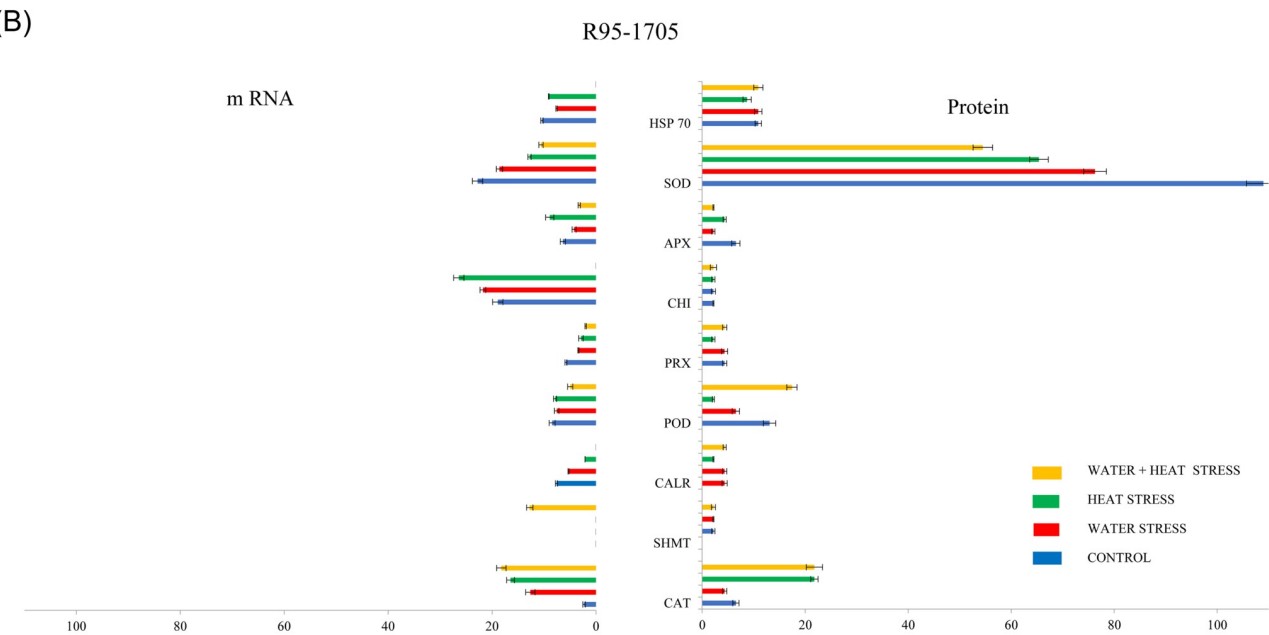

**Fig 9. Changes in transcript levels and their protein abundance in response to various stresses in in soybean leaf of (A): PI-471938 and (B): R95-1795 soybean cultivars.** Relative mRNA abundances were normalized against actin gene abundance. Stress 1: Water Stress; Stress 2: Heat Stress, and Stress 3: Water + Heat stress respectively. Data was normalized using the Actin gene Ct value, and extent of change was calculated using the Ct value of the calibrator (control samples -no stress treatment) using the formula 2-ΔΔCt.

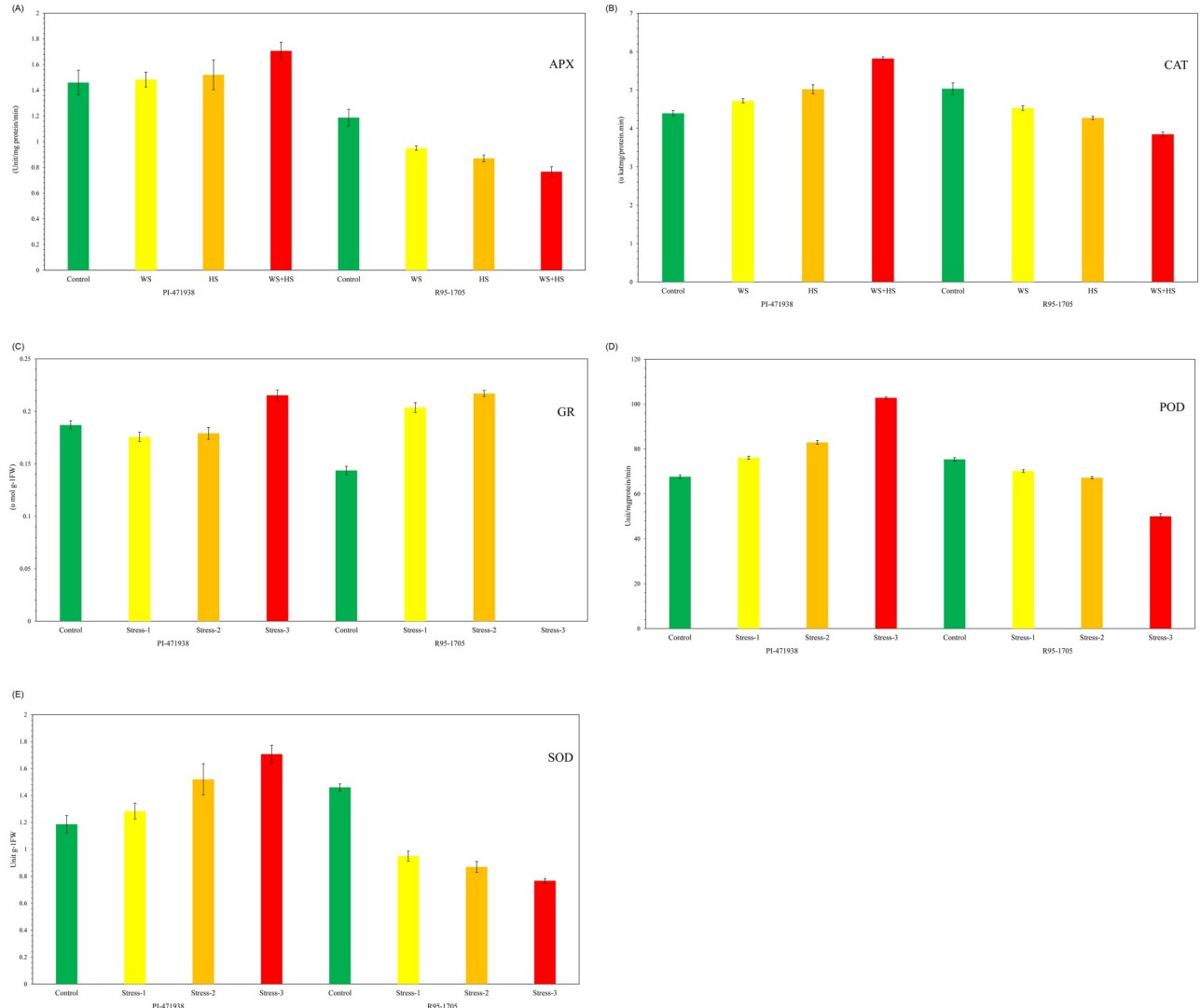

**Fig 10. Enzyme activities under water stress, heat stress and the combined stresses.** The activity is measured in terms of fold change over the control. C: Control; WS: Water stress; HS: Heat stress; WS+HS: Water and Heat stress.

temperatures was closely associated with decrease in soil and plant water status. In both cultivars, the impact of water stress was several folds greater than the effect of high temperature, when these stresses are applied independently. WS, HS, WS+HS reduced stomatal conductance which is caused by leaf water potential via transpiration rate alterations in both cultivars. Photosynthesis is among the primary processes affected by WS+HS [56]. No significant reduction in photosynthesis was observed during HS alone, whereas the consequences of water deficiency due to WS or WS+HS have more significant impact on altering photosynthetic machinery in both cultivars. Both stomatal conductance ($g_s$) and transpiration (Tr) were severely reduced under water stress, accompanied by reduction in the soil and plant water status. Plants partially close stomata to reduce the net transpiration under water stress, which might lead to the observed severe decrease in the photosynthesis ($P_{net}$) [57]. However, the combined effect of two or more stress factors including high temperature has been reported to be more deleterious than the effect of a single stress factor on plant growth [58]. Earlier studies

showed reduction in photosynthesis and dry matter productions due to water stress and high temperature in soybean [14, 59, 60]. Compared to the control, despite the lesser reduction in $P_{net}$ and leaf water content, cultivar R95–1705 showed greater decrease in the total biomass than in PI-471938.

Changes in proteins and anti-oxidative enzymes implicate the morphological and physiological adaptation in plants to stress [61, 62]. Hence, the proteomic studies were carried out to evaluate the relative abundance of proteins and correlate with similar enzyme activities using two contrasting soybean cultivars in response to drought stress. Furthermore, we studied the effect of individual and combined stresses to water and heat stress and to determine the molecular mechanism for cross-tolerance between multiple stresses. Genetic variation was observed for responses to two stresses. PI-471938 cultivar was affected to WS displaying a 25% reduction in proteins, whereas, R95-1705 exhibited a greater reduction in response to HS in proteins. Reduction in proteins was reported in wheat heat-sensitive cultivar compared to tolerant wheat [63]. However, when combination of WS+HS was administered, a similar reduction rate in the number of proteins was observed in both cultivars. Identification of protein isoforms with the same accession numbers spotted at multiple locations may be present due to alternative splicing, polymorphism, and post-translation modifications and they add to the proteome complexity [64]. Often the protein isoforms differ in their cellular concentration and can be used as biomarkers [65].

**Dynamics of stress-responsive proteins indicates PI-471938 cultivar, a heat-tolerant.** Cluster I proteins that showed high abundance to one or more stresses, in contrast to cluster II proteins which showed low abundance to one or more stresses are represented in almost identical numbers in both cultivars, and the majority are metabolism proteins. Among cluster II proteins, a greater reduction in abundance of heat response proteins and photosynthesis related proteins were observed in the R95-1705 cultivar. In R95-1705, eleven proteins were reduced in abundance to all three types of stresses, in particular, affecting more heat response proteins followed by redox proteins. In PI-471938 cultivar, seven proteins showed reduced abundance in all three stress types, primarily involving in metabolism and redox proteins. Cluster III proteins were all involved in metabolism (spot # 18, 20, and 21) showed no response to either water, or, heat, or combined stresses in R95-1705, while they were low in abundance in PI-471938 cultivar.

Cluster IV proteins that displayed mixed responses included twelve proteins from PI-471938 cultivar and six proteins from R95-1705 cultivar. In PI-471938 cultivar, ten proteins showed potential cross-tolerance, five with water and the combined stresses, and five with heat stress and the combined stress, the majority of them are heat response proteins. Heat response proteins that were overly abundant to combined stress in PI-471938 were also more abundant to HS, while two of them were low abundant to WS, suggesting that the cultivar is more responsive and tolerant to HS and to combined HS+WS stress than WS alone.

The stress memory to one stress may prevent damages accruing from other stresses [66]. Plants use stress memory to stabilize performance when exposed to infrequent environmental changes and increase resilience. Our studies revealed that, in PI-471938 cultivar, the majority of proteins involved in heat response and redox had shown either water or heat stress memory. Whereas, the majority of proteins involved in metabolism and photosynthesis have shown either water or heat stress memory, and redox proteins showed heat stress memory in R95-1705 cultivar. Accumulation of transcription factors or proteins facilitates a fast response to repeated stress exposure [67].

**Effect of heat stress showed high abundance of the heat-responsive proteins in PI-471938 cultivar.** The majority of heat response proteins-heat shock protein 70, 22 kDa heat shock protein, 17.7 kDa class 1 small heat shock protein, and 17.6 kDa class 1 heat shock

protein, were consistently high in abundance to heat stress in PI-471938 cultivar, but were low in R95-1705 cultivar. HSP70 is one of the most critical proteins in the response to heat stress, and studies show that this protein is directly linked to the thermotolerance of the plant [68]. HSPs and small HSP make up molecular chaperones and involved in protein folding, prevention of protein aggregation, translocation of proteins across membranes, targeting proteins towards degradation, and regulation of translation initiation, thus the renaturation of stress-damaged proteins protecting cells against the effects of stress [69]. Some HSPs are also involved in transcriptional activation of additional small HSP promoters [70]. Plants up-regulate HSPs, in particular HSP-70, are more tolerant of heat stress [71].

**Effect of water stress showed a high abundance of proteins involved in metabolism in R95-1705 cultivar.** Out of the total 12 proteins involved in the metabolic processes, four proteins were relatively more abundant with 5 proteins maintaining their abundance in R95-1705, only two proteins were more abundant in PI-471938, and two maintained their abundance.

Alanine aminotransferase 2 was the only protein that was more abundant in response to WS in both cultivars. The protein is found in peroxisomes and involved in the degradation of amino acids in plant cells [72]. The abundance of these enzymes indicates that the amino acid metabolism and the synthesis of other metabolites derived from amino acids are well maintained under drought stress [73]. Acid phosphatase was overexpressed to WS in R95-1705 and remained unchanged in PI-471938 cultivar. The role of acid phosphatase is essential in maintaining metabolic homeostasis during drought stress [74].

Serine hydroxymethyltransferase 5, a glycolytic protein involved in photorespiration was not expressed in the control plant however was induced to combination of WS+HS stresses in R95-1705 cultivar. These proteins are directly involved in initiation and elongation of the newly growing peptide chains, indicating severely reduced synthetic protein capacity under drought. Serine hydroxymethyltransferase (SHMT; EC 2.1.2.1) is involved in the photorespiratory pathway of oxygenic photosynthetic organisms [75]. SHMT, a pyridoxal phosphate-dependent enzyme, plays a pivotal role in cellular one-carbon pathways by catalyzing the interconversion of L-serine to glycine and tetrahydrofolate to 5,10-methylenetetrahydrofolate for synthesis of nucleic acids, and proteins [76].

Translation elongation factor Tu and Nucleoside diphosphate kinase (NDPK) is more abundant to WS in R95-1705 cultivar. Translation elongation factor Tu is a GTPase that is responsible for delivering amino-acylated tRNAs to the ribosome during translation [77]. NDPK is found in the matrix and the inner membrane of mitochondria, which regulates the cellular physiology, is known to interact with heat shock proteins [78].

**Combined water and heat stress significantly altered metabolism, redox and photosynthesis-related proteins in both cultivars.** Eleven proteins (#1, 3, 8, 9, 10, 12, 18, 25, 27, 28, and 36) were more abundant in PI-471938 cultivar, with the majority involved in heat response and photosynthesis. Among these, five proteins were in low abundance to WS when applied independently. Twelve proteins (#3, 5, 7, 8, 9, 12, 14, 15, 22, 23, 24, and 31) were more abundant to combined stress in R95-1705 cultivar. The majority were involved in metabolism, out of which, three proteins were low in abundance to WS and two proteins to HS. Soybean cultivar PI-471938, which exhibits a slow-wilting phenotype under water-deficit conditions, has proven to be a good genetic resource in developing drought-resistant progeny [79].

**MED37C as a candidate protein.** Our systems bioinformatics approaches studies showed that more proteins were expressed in higher abundance to combined stresses in PI-471938 cultivar and having more promotive interactions with other proteins, and in particular the proteins associated with metabolism, and photosynthesis, leads to higher performance of the plant to the multiple stresses. Protein MED37C (P26413), identified in our studies, was also observed to have the good potential interacting partners in *Arabidopsis thaliana and* it has

been shown to have two promotive interactors in both cultivars. In R95-1705 cultivar, it has more inhibitory interactors (five), suggesting that the effect of the protein interaction is more significant in PI-471938 than in R95-1705 cultivar in response to combined stresses. Therefore, we contemplate that the protein might also have potential interacting partners in soybean. This candidate protein would be of potential interest as it is a probable mediator of RNA polymerase transcription II associated with heat shock proteins [80].

**Co-relation of mRNA expression for selected stress-responsive proteins.** The protein abundance levels as a result of differential expression of mRNA transcript levels can often be correlated with their protein expression. Although most of the mRNA expression levels in this study exhibited expression trends that were matching with corresponding protein abundance levels, there is a mixed response in the expression and accumulation pattern of some mRNA and the corresponding protein abundance profiles. In PI-471938 cultivar, expression of serine and heat shock protein at both transcript level and protein abundance corresponded with protein abundance and were increased to all stress treatments. However, the expressions of peroxidase, chalcone flavone isomerase, ascorbate perodxidase, catalase, calireticulin, peroxiredoxin, and superoxide dismutase transcripts were in contrast to their protein abundance. There is conflict as the protein abundance is not correlated with the expression of corresponding transcript levels. In cultivar R95-1705, the expression levels of transcript follow their relative abundance in proteins. Broadly, catalase, serine, chalcone flavone isomerase were up-regulated at transcript level and showed relatively high abundance of proteins in response to the stress treatments, while calireticulin, peroxidase, peroxiredoxin, ascorbate peroxidase, superoxide dismutase, and heat shock protein were low at both transcript and protein levels. Reports indicate a poor correlation between mRNA and protein abundances in the cell and depend on various biological and technical factors [81]. The association may not be as similar because the mRNA transcription is relatively lower than protein translation [82]. Protein levels are more conserved than mRNA levels, and their turnover is probably influencing the correlation between mRNA and protein abundances to a higher degree [83].

**Elevated activities in antioxidant enzymes in PI-471938 cultivar enhances the tolerance to ROS production and builds homeostasis.** Antioxidant enzyme activities displayed significantly higher levels in PI-471938 cultivar to all stress treatments, despite their activities low under control conditions when compared with those of R95-1705 cultivar. SOD is considered first-line defense against toxic effects of elevated ROS and the increase in SOD in PI-471938 cultivar to all three stresses would play a significant role in ROS scavenging in plants and is considered as the first line of defense against the toxic effects of elevated ROS levels [84]. SOD catalyzes the dismutation of superoxide radicals to $H_2O_2$ and $O_2$. The increase of SOD activity might be the reason for enhanced $O_2$ generation, as a result of electron leakage from the electron transport chains to molecular oxygen [85]. Water stress or combined WS+HS reportedly induce oxidative stress in plants [86]. POD is the primary enzyme and the increase in its activity detoxifies $H_2O_2$ in chloroplast and the cytosol during oxidative stress [87]. Both POD and CAT constitute a main $H_2O_2$ scavenging system during oxidative stress induced by WS or HS or both [88]. Similar observations of higher POD and CAT activities were reported in tolerant genotypes implicating their role in developing resistance to one or more stresses when compared to the decrease in these enzyme levels in susceptible genotypes [89].

APX activity and GR were elevated in PI-471938 cultivar to all stresses, while their levels GR has not detected in R95-1705 cultivar when treated with combined stresses. Overexpression of APX seems to play a key role in regulating of $H_2O_2$ levels in plant cells by preventing $H_2O_2$ from reaching the nuclei from cytosol, inhibiting lipid peroxidation and protein oxidation and thereby making the cultivar more tolerant [90]. Increased APX indicates that the PI-471938 cultivar can restore the oxidation levels and sustain the plant during the stress. GR

keeps GSH/GSSG ratio favorable for ascorbic acid reduction. The GR activity in PI-471938 cultivar preserved GSH/GSSH ratio due to lower incidence of oxidative damage. In contrast, the absence of GR activity in R95-1705 cultivar in response to combined stress suggests impairment of GSH recycling due to enhanced ROS accumulation [91].

## Conclusions

This study provides insights into proteome and enzyme responses to multiple stresses occurring simultaneously in the field conditions of soybean. Our results showed that concurrent stresses alter physiological, proteome and enzymes indifferent to individual stress. Cultivar PI-471938 maintained total biomass to all stresses, compared to R95-1705 cultivar. The degree of genetic diversity was observed between two cultivars in their protein abundance when subjected to various types of stresses. Several proteins involved in metabolism, response to heat and photosynthesis have shown significant cross-tolerance mechanism. Cross-tolerance was evident in R95-1705 cultivar among heat responsive proteins, photosynthesis, metabolism and redox proteins that were high in abundance to heat stress as well as to the combined heat and water stress in the PI-471938 cultivar suggesting the cultivar as relatively heat tolerant. Both sets of proteins were adversely affected when treated with heat stress alone. However, heat stress alone enhanced redox related proteins in R95-1705 cultivar.

Elevated activities in antioxidant enzymes, such as increased APX, indicate that the PI-471938 cultivar has the ability to restore the oxidation levels and sustain the plant during the stress. Proteins were elevated in high abundance to combined stress in PI-471938 demonstrated more promotive interactions associated with metabolism, photosynthesis leading to continued resistance to both types of stress. Protein MED37C, a probable mediator of RNA polymerase transcription II yielded potential protein interactors partners in *Arabidopsis* and our studies documents the significant impact of the protein in PI-471938 cultivar. Levels of protein expression and transcripts correlate with the regulation at transcription and post transcription levels. Furthermore, the milder stress on small scale can mitigate the detrimental effect of extreme conditions.

## Supporting information

**S1 Fig. Close up view of selected proteins.**
(PPT)

**S2 Fig. Relative protein abundance to water, heat, and both water and heat stresses in soybean cultivars.**
(PPTX)

**S3 Fig. Protein abundance in stress treatments relative to control determined by MapMan.**
(PPTX)

**S1 Table. List of primers used for real-time RT-qPCR.**
(PDF)

**S2 Table. Two-DE Reproducibility of two-dimensional electrophoresis gels.**
(PDF)

**S3 Table. Promotive and inhibitive effect of stress responsive proteins.**
(PDF)

**S4 Table. Promotive and inhibitive effect of stress responsive proteins showing high abundance in response to combined water and heat stresses Cultivar PI-471938.**
(PDF)

**S5 Table. Promotive and inhibitive effect of stress responsive proteins showing high abundance in response to combined water and heat stresses Cultivar R95–1705.**
(DOCX)

**S1 Data.**
(PDF)

**S1 Graphical Abstract.**
(PDF)

## Acknowledgments

The soybean lines were generously supplied by Dr. Thomas E. Carter, Jr., USDA-ARS Soybean & Nitrogen Fixation Unit, Raleigh, NC and Dr. Pengyin Chen, University of Missouri, Columbia, MO. Drs. Virginia Gottschalk, Mohammad Akbari, Ms. Tiffany Boyunt, Catlyn Eliason and Sara Bergin reviewed or assisted in preparing figures.

## Author Contributions

**Conceptualization:** Ramesh Katam, Kambham Raja Reddy.

**Formal analysis:** Sedigheh Shokri, Nitya Murthy, Shardendu K. Singh, Prashanth Suravajhala, Mudassar Nawaz Khan, Katsumi Sakata.

**Investigation:** Ramesh Katam, Kambham Raja Reddy.

**Methodology:** Ramesh Katam, Nitya Murthy, Shardendu K. Singh, Prashanth Suravajhala, Kambham Raja Reddy.

**Project administration:** Kambham Raja Reddy.

**Resources:** Mahya Bahmani.

**Software:** Prashanth Suravajhala, Mudassar Nawaz Khan, Katsumi Sakata.

**Supervision:** Kambham Raja Reddy.

**Validation:** Sedigheh Shokri, Prashanth Suravajhala, Katsumi Sakata, Kambham Raja Reddy.

**Writing – original draft:** Ramesh Katam.

**Writing – review & editing:** Ramesh Katam, Sedigheh Shokri, Nitya Murthy, Prashanth Suravajhala, Mudassar Nawaz Khan, Mahya Bahmani, Katsumi Sakata, Kambham Raja Reddy.

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
