## [Decision Letter · Decision Letter 0]

31 Dec 2019

PONE-D-19-32175

Proteomics,  Physiological, and Biochemical Analysis of Tolerance Mechanisms in Response to Heat and Water Stresses in Soybean

PLOS ONE

Dear Dr katam,

Thank you for submitting your manuscript to PLOS ONE. After careful consideration, we feel that it has merit but does not fully meet PLOS ONE’s publication criteria as it currently stands. Therefore, we invite you to submit a revised version of the manuscript that addresses the points raised during the review process.

We would appreciate receiving your revised manuscript by Feb 14 2020 11:59PM. To enhance the reproducibility of your results, we recommend that if applicable you deposit your laboratory protocols in protocols.io, where a protocol can be assigned its own identifier (DOI) such that it can be cited independently in the future. For instructions see: http://journals.plos.org/plosone/s/submission-guidelines#loc-laboratory-protocols

We look forward to receiving your revised manuscript.

Kind regards,

Haitao Shi

Academic Editor

PLOS ONE

Journal Requirements:

2. Please include your tables as part of your main manuscript and remove the individual files. Please note that supplementary tables should remain as separate "supporting information" files

4. Please upload a copy of Supporting Information Table V which you refer to in your text on page 23.

Reviewers' comments:

Reviewer's Responses to Questions

**Comments to the Author**

1. Is the manuscript technically sound, and do the data support the conclusions?

Reviewer #1: Yes

2. Has the statistical analysis been performed appropriately and rigorously? 

Reviewer #1: No

3. Have the authors made all data underlying the findings in their manuscript fully available?

Reviewer #1: Yes

4. Is the manuscript presented in an intelligible fashion and written in standard English?

Reviewer #1: Yes

5. Review Comments to the Author

Reviewer #1: Remarks:

> The Authors tested two soybean cultivars. One of the cultivars is characterized as „slow wilting and high yielding” and the second one as “high in protein concentration and moderate yield potential’. I am not sure if they differ from each other as regards the phenotype connected with the studied stresses. I suggest giving information on their susceptibility to the studied stresses.

> Besides, the Authors use different names for the tested two cultivars. I don't understand why it is so. It is absolutely necessary to sort it out. The cultivar PI 471938 is also called: Cultivar#2; PI; 471938; PI 373819; PI-371938; DT, whilst the cultivar R95 – 1795 is also named: Cultivar#1; R-95; R95-1705; R95; R 95-1705; DS.

Material and Methods

> line 147: Physiological measurements - how many replications were used for the analysis?

> line 162: Statistical analysis – this subsection should be transferred to the end of the section Material and Methods

> line 222: Quantitative Real-Time Polymerase Chain Reaction (RT-q PCR) Analysis: What plant material and in what amount was used for RNA isolation? What method was used for RNA isolation? How was the reverse transcription performed? Please, give the PCR parameters (annealing temperature and time). Method for identifying the RT-qPCR products; method for evaluating the quantity of transcript. What was the reference gene, if any? Were there negative controls? Was the effectiveness of RT-PCR amplification assessed using standard gel electrophoresis? The expression of which genes was studied? Please, name these genes.

> line 228: Enzyme assay: No description of how the samples were prepared for determination, and what plant material was used for that purpose. How much plant material was used for determination? How many biological replicates were made? Lack of statistics.

Results

> Abbreviations, when used for the first time in Material and Methods, should be explained. Pnet, Tr, TBM, Fv’/Fm’. How was TBM determined? – no description in Material and Methods.

> lines 248-250, and Table 1 – please check the font

> line 275; correlation coefficient: please check it

> line 308: “mixed response” please explain what it means

> line 316: “protein 7 and 12 …..were not present” – it is not a correct interpretation, this would mean that in the control no proteins undergo expression, while the level of the tested proteins shows an increase or decrease in relation to the control point (see the scale) which was 1.

> line 344: Supplementary Table V is not present in the manuscript

> line 358: Figure 9b is not present in the manuscript

> line 371: “significantly higher” – in order to conclude that something is significantly higher, ANOVA should have been performed

Figure and Table Legends:

Figure titles are insufficient. There is a lack of figure legends, so in consequence the interpretation of the presented results is complicated or even impossible.

Please, check the title of Figure 6, in the title of Figure 7 – “what does A:” mean?

Figure 9 title: “transcript expression’ is not a proper term: I suggest to use “transcript accumulation” or “transcript level”

Figures

Figure 1: needs to be improved, please check the names of the cultivars

Figure 2: Please, explain in the legend what 3-1-1 etc. mean; please, check the names of the cultivars

Figure 3: is not clear, please explain what Figure 3b expresses

Figure 5: needs to be improved

Figure 6: Please, give information about the treatment under the column for both the cultivars. Please sign the A) and B) parts to the figure. Please name the part of the figure with the numbers on the left and right sides; the clusters are not visible.

Figure 7: It should be improved. It is improper and unnecessary to introduce additional determinations for cultivars in that part of the manuscript. The figure is not divided into a, b and c, the references to which are in the text and the legend. Please, explain what the numbers on the graph mean. The lower part of the figure is a chart which has unfortunately been constructed in a way which makes it impossible to see all the data.

Figure 9: There is a lack of the part connected with the cultivar R95-1705. Please, check the name of the proteins – please, see also Supplementary Table I. Relative mRNA abundance – no description of the axis. Protein – no description of the axis. In general, a lack of statistics. No explanation of what the figure is about. Proteins – how ‘protein abundance’ was calculated?

Figure 10. Lack of statistics. Lack of description of the Y axis in the figures.

Supplementary Figure I; Lack of a legend to the figure.

Supplementary Figure II; Lack of a legend to the figure. What do the numbers on the X axis mean? Lack of the Y axis unit.

Supplementary Figure III: illegible – needs to be improved.

Tables

Supplementary Table I: Please, check the names of the proteins; please, show the direction of the primer sequence.

Supplementary Table II: Please, check the title of the table, the names of the cultivars. Generally, the table needs to be improved.

Supplementary Table III: Generally, the table needs to be improved. ‘SI. No.’ – please explain what it means.

References

Please, correct it; it should be prepared according to the manufacturer’s instructions.

6. PLOS authors have the option to publish the peer review history of their article (what does this mean?). If published, this will include your full peer review and any attached files.

Reviewer #1: No

---

## [Author Response · Author response to Decision Letter 0]

14 May 2020

Thank you very much for considering the manuscript suitable for publication in PLoS One journal. We included the following files; Rebuttal, Manuscript, and Revised Manuscript With Track Changes for further review and consideration. Please note tht our responses are highlighted in BLUE. 

We look forward to receiving your decision on this manuscript.

Kind regards,

Ramesh Katam

Response: Entire manuscript was revised to meet PLOS ONE’s style requirements.

2. Please include your tables as part of your main manuscript and remove the individual files. Please note that supplementary tables should remain as separate "supporting information" files

Response: Tables were included as part of main manuscript and removed the individual files. Supplementary tables were included as “supporting information’ files. 

Response: Original gels uncropped were included as Supporting Information. 

4. Please upload a copy of Supporting Information Table V which you refer to in your text on page 23.

Response: Supporting Information Table V is included. 

Reviewers' comments:

Reviewer's Responses to Questions

Comments to the Author

1. Is the manuscript technically sound, and do the data support the conclusions?

Reviewer #1: Yes

Response: Thank you very much for the comments on the quality of the manuscript data. 

2. Has the statistical analysis been performed appropriately and rigorously?

Reviewer #1: No

Response: Statistical analysis has been performed for all the figures and for the data of the manuscript and revised as needed. 

3. Have the authors made all data underlying the findings in their manuscript fully available?

Reviewer #1: Yes

Response: Thank you for your comment. 

4. Is the manuscript presented in an intelligible fashion and written in standard English?

Reviewer #1: Yes

Response: Thank you for your comment. 

5. Review Comments to the Author

Reviewer #1: Remarks:

> The Authors tested two soybean cultivars. One of the cultivars is characterized as „slow wilting and high yielding” and the second one as “high in protein concentration and moderate yield potential’. I am not sure if they differ from each other as regards the phenotype connected with the studied stresses. I suggest giving information on their susceptibility to the studied stresses.

Response: These are newly developed cultivars and have not been fully studied for their response to drought. However, the PI genotype has been tested for heat-tolerance and showed those tolerance traits. Additional information on these cultivars are provided in the revised manuscript. 

> Besides, the Authors use different names for the tested two cultivars. I don't understand why it is so. It is absolutely necessary to sort it out. The cultivar PI 471938 is also called: Cultivar#2; PI; 471938; PI 373819; PI-371938; DT, whilst the cultivar R95 – 1795 is also named: Cultivar#1; R-95; R95-1705; R95; R 95-1705; DS.

Response: The cultivar names were revised and are consistent in the entire manuscript. 

Material and Methods

> line 147: Physiological measurements - how many replications were used for the analysis? 

Response: we used 6 replications for all physiological measurements and 10 replications for total biomass. Accordingly, the materials and methods section was updated. 

> line 162: Statistical analysis – this subsection should be transferred to the end of the section Material and Methods

Response: The Statistical analysis subsection was transferred to the end of section Materials and Methods. 

> line 222: Quantitative Real-Time Polymerase Chain Reaction (RT-q PCR) Analysis: What plant material and in what amount was used for RNA isolation? What method was used for RNA isolation? How was the reverse transcription performed? Please, give the PCR parameters (annealing temperature and time). Method for identifying the RT-qPCR products; method for evaluating the quantity of transcript. What was the reference gene, if any? Were there negative controls? Was the effectiveness of RT-PCR amplification assessed using standard gel electrophoresis? The expression of which genes was studied? Please, name these genes.

Response: Thank you for your comment. Revised and included the details about plant material, quantity, RNA isolation method, RT PCR parameters, reference gene, list of genes quantified, and the method of product validation. 

> line 228: Enzyme assay: No description of how the samples were prepared for determination, and what plant material was used for that purpose. How much plant material was used for determination? How many biological replicates were made? Lack of statistics. 

Response: Detailed description of sample preparation, plant material used, including biological replicates was explained in methods section. Figures associated with enzyme assay was prepared using statistics. 

Results

> Abbreviations, when used for the first time in Material and Methods, should be explained. Pnet, Tr, TBM, Fv’/Fm’. How was TBM determined? – no description in Material and Methods. 

Response: Thanks. We have explained in the materials and methods section and also in the Table.

> lines 248-250, and Table 1 – please check the font.

Response: Fonts were checked and revised.

> line 275; correlation coefficient: please check it 

Response: Checked and corrected.

> line 308: “mixed response” please explain what it means

Response: Revised. 

> line 316: “protein 7 and 12 …..were not present” – it is not a correct interpretation, this would mean that in the control no proteins undergo expression, while the level of the tested proteins shows an increase or decrease in relation to the control point (see the scale) which was 1.

Response: Revised and deleted the statement. 

> line 344: Supplementary Table V is not present in the manuscript

Response: Supplementary Table V is provided.

> line 358: Figure 9b is not present in the manuscript

Response: Figure 9 b which is related to cultivar R95-1705 is included.

> line 371: “significantly higher” – in order to conclude that something is significantly higher, ANOVA should have been performed.

Response: Performed ANOVA and verified for significant differences and the data is provided as supplementary document. 

Figure and Table Legends:

Figure titles are insufficient. There is a lack of figure legends, so in consequence the interpretation of the presented results is complicated or even impossible.

Response: Figure titles are described in detail with the legends for better visual expression. 

Please, check the title of Figure 6, in the title of Figure 7 – “what does A:” mean?

Response: Revised and given the cultivar names. 

Figure 9 title: “transcript expression’ is not a proper term: I suggest to use “transcript accumulation” or “transcript level”

Response: Revised to “transcript level”

Figures

Figure 1: needs to be improved, please check the names of the cultivars.

Response: Figure 1 is improved and corrected the cultivar names.

Figure 2: Please, explain in the legend what 3-1-1 etc. mean; please, check the names of the cultivars

Response: Revised the names of cultivars and the treatments.

Figure 3: is not clear, please explain what Figure 3b expresses

Response: Information provided.

Figure 5: needs to be improved

Response: Improved Figure 5

Figure 6: Please, give information about the treatment under the column for both the cultivars. Please sign the A) and B) parts to the figure. Please name the part of the figure with the numbers on the left and right sides; the clusters are redrawn to improve the visibility.

Response: The clusters are redrawn to improve the visibility. The numbers were added on left and right side of the cluster diagrams. Figure part a and b were assigned and labeled the cultivar name. Figure was numbered on left and right sides with the numbers and uniport IDs. 

Figure 7: It should be improved. It is improper and unnecessary to introduce additional determinations for cultivars in that part of the manuscript. The figure is not divided into a, b and c, the references to which are in the text and the legend. Please, explain what the numbers on the graph mean. The lower part of the figure is a chart which has unfortunately been constructed in a way which makes it impossible to see all the data 

Response: The figure quality is improved. The figure is divided into a, b and c in reference to the information in the text. The numbers in the graph indicate number of proteins related to the biological functions affected. The graph was re-constructed to show more details. 

Figure 9: There is a lack of the part connected with the cultivar R95-1705. Please, check the name of the proteins – please, see also Supplementary Table I. Relative mRNA abundance – no description of the axis. Protein – no description of the axis. In general, a lack of statistics. No explanation of what the figure is about. Proteins – how ‘protein abundance’ was calculated?

Response: Figure 9 (a & b) is about the relation to transcript level and corresponding protein relative abundance. The protein names were revised and also corrected in Supplementary Table I. X and Y axis for mRNA and protein were described in the table. Protein abundance was calculated using PD Quest software (Bio Rad) which uses the algorithm of relative protein spot density. In this method, the raw quantity of each spot in a member gel is divided by the total intensity value of all the pixels in the image. 

Figure 10. Lack of statistics. Lack of description of the Y axis in the figures.

Response: The figure is redrawn using Statistics measurements, and Y axis is described.

Supplementary Figure I; Lack of a legend to the figure.

Response: Revised.

Supplementary Figure II; Lack of a legend to the figure. What do the numbers on the X axis mean? Lack of the Y axis unit.

Response: Revised.

Supplementary Figure III: illegible – needs to be improved.

Response: Revised and improved, enlarged to more legible and split into 3 parts.

Tables

Supplementary Table I: Please, check the names of the proteins; please, show the direction of the primer sequence.

Response: Names of the proteins, and primer sequence directions were corrected and revised.

Supplementary Table II: Please, check the title of the table, the names of the cultivars. Generally, the table needs to be improved.

Response: Revised.

Supplementary Table III: Generally, the table needs to be improved. ‘SI. No.’ – please explain what it means.

Response: Revised and improved.

References

Please, correct it; it should be prepared according to the manufacturer’s instructions.

Response: References were prepared according to the manufacturer’s instructions.

---

## [Editor Report · Decision Letter 1]

15 May 2020

Proteomics,  Physiological, and Biochemical Analysis of Cross Tolerance Mechanisms in Response to Heat and Water Stresses in Soybean

PONE-D-19-32175R1

Dear Dr. katam,

We are pleased to inform you that your manuscript has been judged scientifically suitable for publication and will be formally accepted for publication once it complies with all outstanding technical requirements.

With kind regards,

Haitao Shi

Academic Editor

PLOS ONE
---

## [Editor Report · Acceptance letter]

20 May 2020

PONE-D-19-32175R1 

Proteomics,  Physiological, and Biochemical Analysis of Cross Tolerance Mechanisms in Response to Heat and Water Stresses in Soybean 

Dear Dr. Katam:

I am pleased to inform you that your manuscript has been deemed suitable for publication in PLOS ONE. Congratulations! Your manuscript is now with our production department. 

With kind regards,

on behalf of

Dr. Haitao Shi 

Academic Editor

PLOS ONE